# Sharpness-Aware Minimization in Logit Space Efficiently Enhances Direct Preference Optimization

**Haocheng Luo**[1]    **Zehang Deng**[2]    **Thanh-Toan Do**[1]    **Mehrtash Harandi**[1]
**Dinh Phung**[1]    **Trung Le**[1]
[1]Monash University, Australia
[2]Swinburne University of Technology, Australia
{haocheng.luo, toan.do, mehrtash.harandi, dinh.phung, trunglm}@monash.edu
zehangdeng@swin.edu.au

## Abstract

Direct Preference Optimization (DPO) has emerged as a popular algorithm for aligning pretrained large language models with human preferences, owing to its simplicity and training stability. However, DPO suffers from the recently identified *squeezing effect* (also known as *likelihood displacement*), where the probability of preferred responses decreases unintentionally during training. To understand and mitigate this phenomenon, we develop a theoretical framework that models the coordinate-wise dynamics in logit space. Our analysis reveals that negative-gradient updates cause residuals to expand rapidly along high-curvature directions, which underlies the squeezing effect, whereas Sharpness-Aware Minimization (SAM) can suppress this behavior through its curvature-regularization effect. Building on this insight, we investigate *logits-SAM*, a computationally efficient variant that perturbs only the output layer with negligible overhead. Extensive experiments on Pythia-2.8B, Mistral-7B, and Gemma-2B-IT across multiple datasets and benchmarks demonstrate that logits-SAM consistently improves the effectiveness of DPO and integrates seamlessly with other DPO variants. Code is available at https://github.com/RitianLuo/logits-sam-dpo.

## 1 Introduction

Reinforcement learning from human feedback (RLHF) (Christiano et al., 2017; Stiennon et al., 2020; Ouyang et al., 2022) is a crucial technique for aligning pretrained large language models (LLMs) with human preferences to ensure helpfulness, harmlessness, and safety (Bai et al., 2022; Dai et al., 2023). Its pipeline typically comprises three stages: supervised fine-tuning (SFT), reward modeling, and policy optimization. Classical policy optimization methods such as Proximal Policy Optimization (PPO) (Schulman et al., 2017), while widely used for their effectiveness, depend heavily on the quality of the learned reward model, rendering training complex and often unstable. Direct Preference Optimization (DPO) (Rafailov et al., 2024b) is a recently proposed and promising offline alternative that, by reparameterizing the implicit reward and optimizing a closed-form objective on preference data, trains the policy directly without explicitly fitting a reward model. DPO has gained traction due to its algorithmic simplicity and training stability.

Despite DPO and its many variants demonstrating state-of-the-art performance across a range of tasks, several potential issues remain. A particularly important one is the recently identified *squeezing effect* (Ren & Sutherland, 2024) (also known as *likelihood displacement* (Razin et al., 2024)), which describes an unintended decrease in the generation probability of preferred responses during DPO training, contrary to the intended goal of increasing it embodied in the DPO objective. This phenomenon can lead to performance degradation, reduced safety, and even alignment failure (Pal et al., 2024; Yuan et al., 2024; Rafailov et al., 2024a; Tajwar et al., 2024; Pang et al., 2024).

To understand the mechanism behind the squeezing effect and to identify an effective remedy, we develop a theoretical framework that elucidates the learning dynamics in both the parameter space and the logit space. Our analysis shows that the negative-gradient updates induced by the negative objective associated with rejected answers in DPO cause the residual vector to expand rapidly along high-curvature directions, namely along the eigenvectors associated with large eigenvalues of the Hessian, which is the source of the squeezing effect. This raises a natural question: *Can curvature-aware training mitigate this unintended drift?*

We investigate *Sharpness-Aware Minimization* (SAM) (Foret et al., 2021), a bilevel optimization method widely used in supervised learning, and establish its dynamics in both the parameter and logit spaces. Our theory demonstrates that SAM effectively alleviates the squeezing effect through its intrinsic curvature regularization. Guided by these insights, we advocate using *logits-SAM* for DPO training, a computationally efficient variant of SAM that perturbs only the output-layer parameters. Although logits-SAM has been mentioned merely as a byproduct in prior work (Baek et al., 2024; Singh et al., 2025) and often overlooked, our study turns this neglected variant into a practically useful and effective technique by integrating it into DPO, where it efficiently mitigates the squeezing effect and consistently improves performance. To the best of our knowledge, this is the first work to analyze and apply SAM in the context of DPO.

Our contributions are summarized as follows:

- We develop a theoretical framework that connects the parameter space and the logit space through geometric properties, enabling a unified analysis of learning dynamics in both domains. This framework yields unified dynamical equations for gradient descent (GD) and SAM that precisely track coordinate-wise evolution with controlled error terms.

- Our analysis identifies the root cause of the squeezing effect: under negative-gradient updates, residuals expand rapidly along high-curvature directions. We rigorously show that SAM, through its intrinsic curvature regularization, effectively alleviates this phenomenon.

- Bridging theory and practice, we implement an efficient variant, *logits-SAM*, which perturbs only the output-layer parameters. Unlike vanilla SAM, it incurs virtually no additional overhead. Experiments on Pythia-2.8B, Mistral-7B, and Gemma-2B-IT across multiple datasets and benchmarks validate its effectiveness, demonstrating consistent performance gains for DPO and its variants.

## 2 PRELIMINARIES

### 2.1 PREFERENCE OPTIMIZATION

**SFT–RLHF pipeline.** Classical RLHF alignment proceeds in three phases: (i) *supervised fine-tuning* of a base policy on instruction-following data; (ii) *reward modeling* by fitting a scalar reward function on pairwise human preferences; and (iii) *policy optimization* to maximize the learned reward under a KL regularizer toward a reference policy.

**DPO reparameterization.** DPO (Rafailov et al., 2024b) bypasses training an explicit reward model by expressing an *implicit* reward for a policy $\pi_{\boldsymbol{\theta}}$ as a log-likelihood ratio to a fixed reference policy $\pi_{\text{ref}}$ (typically the SFT model):

$$r_{\boldsymbol{\theta}}(\boldsymbol{x}, \boldsymbol{y}) \;=\; \beta \, \log \frac{\pi_{\boldsymbol{\theta}}(\boldsymbol{y} \,|\, \boldsymbol{x})}{\pi_{\text{ref}}(\boldsymbol{y} \,|\, \boldsymbol{x})} \;+\; \beta \log Z(\boldsymbol{x}), \tag{1}$$

where $\beta > 0$ is a temperature and $Z(\boldsymbol{x})$ is a partition term independent of $\boldsymbol{\theta}$. Combining equation 1 with the Bradley–Terry preference model (Bradley & Terry, 1952) $p(\boldsymbol{y}^+ \succ \boldsymbol{y}^- \mid \boldsymbol{x}) = \sigma\big(r_{\boldsymbol{\theta}}(\boldsymbol{x}, \boldsymbol{y}^+) - r_{\boldsymbol{\theta}}(\boldsymbol{x}, \boldsymbol{y}^-)\big)$ yields the standard DPO objective, optimized over a dataset $\mathcal{D} = \{(\boldsymbol{x}, \boldsymbol{y}^+, \boldsymbol{y}^-)\}$ of preferred/dispreferred pairs:

$$\mathcal{L}_{\text{DPO}}(\pi_{\boldsymbol{\theta}}; \pi_{\text{ref}}) = - \, \mathbb{E}_{(\boldsymbol{x}, \boldsymbol{y}^+, \boldsymbol{y}^-) \sim \mathcal{D}} \left[ \log \sigma \left( \beta \log \frac{\pi_{\boldsymbol{\theta}}(\boldsymbol{y}^+ \mid \boldsymbol{x})}{\pi_{\text{ref}}(\boldsymbol{y}^+ \mid \boldsymbol{x})} - \beta \log \frac{\pi_{\boldsymbol{\theta}}(\boldsymbol{y}^- \mid \boldsymbol{x})}{\pi_{\text{ref}}(\boldsymbol{y}^- \mid \boldsymbol{x})} \right) \right], \tag{2}$$

where $\sigma(\cdot)$ is the logistic function.

## 2.2 SHARPNESS-AWARE MINIMIZATION

SAM regularizes training by explicitly penalizing *parameter-space sharpness*: it chooses parameters that minimize the worst-case loss within an $\ell_2$ ball of radius $\rho$ around $\boldsymbol{\theta}$. Concretely, for supervised learning with examples $(\boldsymbol{x}, \boldsymbol{y}) \sim \mathcal{D}$ and per-example loss $f(\boldsymbol{\theta}; \boldsymbol{x}, \boldsymbol{y})$, the SAM objective is

$$\min_{\boldsymbol{\theta}} \quad \mathbb{E}_{(\boldsymbol{x}, \boldsymbol{y}) \sim \mathcal{D}} \left[ \max_{\|\boldsymbol{\epsilon}\|_2 \leq \rho} f(\boldsymbol{\theta} + \boldsymbol{\epsilon}; \boldsymbol{x}, \boldsymbol{y}) \right]. \tag{3}$$

This formulation can be interpreted as a form of *curvature regularization*: by seeking minimizers whose neighborhoods exhibit consistently low loss, SAM favors flatter minima that often correlate with improved generalization. In practice, the inner maximization is approximated to first order by the perturbation $\boldsymbol{\epsilon}^*(\boldsymbol{\theta}) = \rho \nabla_{\boldsymbol{\theta}} f(\boldsymbol{\theta}; \boldsymbol{x}, \boldsymbol{y}) / \|\nabla_{\boldsymbol{\theta}} f(\boldsymbol{\theta}; \boldsymbol{x}, \boldsymbol{y})\|_2$, and one takes a descent step using the gradient at the perturbed point, $\nabla_{\boldsymbol{\theta}} f(\boldsymbol{\theta} + \boldsymbol{\epsilon}^*; \boldsymbol{x}, \boldsymbol{y})$.

## 3 LEARNING DYNAMICS IN LOGIT SPACE

### 3.1 SETTING

We adopt the same theoretical setting as in Ren & Sutherland (2024), namely multiclass logistic classification, where the features of the samples are fixed (also referred to as the kernel regime (Malladi et al., 2023)). In this abstraction, we keep the per-example loss nonnegative, and model the two DPO terms by allowing the *effective* update direction to take either sign: the preferred term ($\boldsymbol{y}^+$) corresponds to standard descent with $\eta > 0$, whereas the rejected term ($\boldsymbol{y}^-$) induces *negative-gradient updates*, which we represent as an update with $\eta < 0$. For analytical convenience, we follow Ren & Sutherland (2024) and encode these negative-gradient updates as gradient descent with a negative effective learning rate, which is equivalent to using a positive learning rate on a negated objective (see Appendix D for a short derivation).

Prior work (Ren & Sutherland, 2024) has shown that the resulting negative-gradient dynamics, including the characteristic squeezing effect, can be faithfully reproduced within this simplified setting. Several phenomena observed in this multiclass logistic regression abstraction also emerge empirically during real LLM fine-tuning. These findings suggest that analyzing DPO through this framework offers a theoretically tractable and practically relevant perspective on the learning behavior of LLMs.

Let $\boldsymbol{x}$ be a training example with one-hot label $\boldsymbol{y} \in \{0, 1\}^V$, $\mathbf{1}^\top \boldsymbol{y} = 1$. In the fixed-feature (kernel) regime, $\phi(\boldsymbol{x}) \in \mathbb{R}^d$ are fixed and

$$\boldsymbol{z}^t = \boldsymbol{W}^t \phi(\boldsymbol{x}) \in \mathbb{R}^V, \qquad \boldsymbol{p}^t = \text{softmax}(\boldsymbol{z}^t), \qquad f(\boldsymbol{z}^t, \boldsymbol{y}) = -\sum_{k=1}^{V} y_k \log p_k^t,$$

where $\boldsymbol{W}^t \in \mathbb{R}^{V \times d}$ are trainable parameters, $\boldsymbol{z}^t$ are the logits. For notational convenience, we write $\phi(\boldsymbol{x})$ as $\phi$. We use $\|\cdot\|$ to denote the $\ell_2$ norm for vectors and the Frobenius norm for matrices. We use $\otimes$ to denote the Kronecker product.

We denote the parameter Hessian by $\boldsymbol{H}_{\boldsymbol{W}}^t := \nabla_{\boldsymbol{W}}^2 f(\boldsymbol{z}^t, \boldsymbol{y}) \in \mathbb{R}^{Vd \times Vd}$, and $\mu := \|\phi\|^2$. We denote the logit gradient by $\boldsymbol{g}^t := \nabla_{\boldsymbol{z}} f(\boldsymbol{z}^t, \boldsymbol{y}) = \boldsymbol{p}^t - \boldsymbol{y} \in \mathbb{R}^V$, and denote the logit Hessian by $\boldsymbol{H}_{\boldsymbol{z}}^t := \nabla_{\boldsymbol{z}}^2 f(\boldsymbol{z}^t, \boldsymbol{y}) \in \mathbb{R}^{V \times V}$.

### 3.2 THEORY

The theoretical results of Ren & Sutherland (2024) demonstrate that the *squeezing effect* arises from negative-gradient updates. Specifically, they show that the probability of the ground-truth label necessarily decreases, while the probability of the model's most confident incorrect class increases. Building on this, we provide a finer-grained analysis of the learning dynamics. We develop a unified modeling framework that tracks the residuals of all classes and characterizes their coordinate-wise evolution with linear convergence up to higher-order remainders. Using this framework, we further establish rigorous conditions under which SAM effectively mitigates the squeezing effect.

For GD, first-order derivatives are sufficient to characterize its dynamics. However, the intrinsic curvature regularization effect of SAM motivates us to further investigate the geometric structure of the parameter space through the Hessian matrix. To this end, we develop a theoretical framework that connects the geometry of the parameter space and the logit space, via the link between the parameter Hessian and the logit Hessian.

**Proposition 3.1** (Geometry of the logit space; simplified version of Proposition B.1). *In coordinates,* $\boldsymbol{H_W} = (\phi\phi^\top) \otimes \boldsymbol{H_z}$. *Thus, if* $\phi \neq \boldsymbol{0}$, *then* $\mathrm{rank}(\boldsymbol{H_W}) = \mathrm{rank}(\boldsymbol{H_z})$. *Moreover, the second-order effect of any parameter perturbation depends only on the induced logits perturbation* $T_\phi(\Delta\boldsymbol{W}) := \Delta\boldsymbol{W}\,\phi$.

This proposition establishes that all second-order effects in the parameter space, whose Hessian $\boldsymbol{H_W}$ lies in $\mathbb{R}^{Vd \times Vd}$, can be equivalently studied through the logit Hessian $\boldsymbol{H_z}$ in $\mathbb{R}^{V \times V}$, thereby greatly simplifying the analysis of second-order dynamics. Next, building on this geometric correspondence, we derive unified dynamical equations for SAM in both the parameter space and the logit space. Unlike prior work, these dynamics simultaneously capture the evolution of all coordinates of the parameters, logits, and residuals, with precise control over the error terms.

**Theorem 3.2** (SAM dynamics in parameter and logit space; informal version of Theorem B.2). *Assume that we conduct the SAM update for* $\boldsymbol{W}$. *Under mild assumptions, there exists a constant* $C > 0$ *such that the following expansions hold with* $O(\eta^2)$ *remainders:*

$$\textbf{(parameters)} \quad \boldsymbol{W}^{t+1} = \boldsymbol{W}^t - \eta\Big(\boldsymbol{g}^t\,\phi^\top + \underbrace{\tilde{\rho}^t\,\boldsymbol{H_z}^t\boldsymbol{g}^t\,\phi^\top}_{\text{SAM's correction}}\Big) + \boldsymbol{R}_{\boldsymbol{W}}^t, \qquad \|\boldsymbol{R}_{\boldsymbol{W}}^t\| \leq C\,\eta^2,$$

$$\textbf{(logits)} \quad \boldsymbol{z}^{t+1} = \boldsymbol{z}^t - \eta\,\mu\Big(\boldsymbol{g}^t + \underbrace{\tilde{\rho}^t\,\boldsymbol{H_z}^t\boldsymbol{g}^t}_{\text{SAM's correction}}\Big) + \boldsymbol{r_z}^t, \qquad \|\boldsymbol{r_z}^t\| \leq C\,\eta^2,$$

$$\textbf{(residuals)} \quad \boldsymbol{g}^{t+1} = \boldsymbol{p}^{t+1} - \boldsymbol{y} = \Big(\boldsymbol{I} - \eta\,\mu\,\boldsymbol{H_z}^t - \underbrace{\eta\,\mu\,\tilde{\rho}^t\,(\boldsymbol{H_z}^t)^2}_{\text{SAM's correction}}\Big)(\boldsymbol{p}^t - \boldsymbol{y}) + \boldsymbol{r_g}^t, \quad \|\boldsymbol{r_g}^t\| \leq C\,\eta^2,$$

*where* $\tilde{\rho}^t := \rho\,\sqrt{\mu}/\|\boldsymbol{g}^t\|$ *is the equivalent perturbation coefficient.*

When $\rho = 0$, the dynamics reduce to standard GD. Expressed through the logit Hessian, both GD and SAM share a unified dynamical structure across parameter, logit, and residual spaces. In both parameter and logit space, GD amounts to scaling by the logit gradient, whereas SAM introduces an additional $\boldsymbol{H_z}$ correction term that can be regarded as a preconditioning matrix, rescaling updates differently along the eigen-directions of $\boldsymbol{H_z}$.

Moreover, the updates of the residual vector under GD and SAM are both preconditioned by $\boldsymbol{H_z}$ (and, for SAM, by $(\boldsymbol{H_z})^2$). This implies that if we choose the eigenvectors of the logit Hessian as a basis, the curvature coupling effects of both the first-order and second-order terms can be unified. To formalize this intuition, we show that $\boldsymbol{g}$ lies precisely in the column space of $\boldsymbol{H_z}$, thus we can select the nonzero eigenvectors of $\boldsymbol{H_z}$ as a basis to obtain the coordinate representation of $\boldsymbol{g}$.

**Proposition 3.3.** $\boldsymbol{H_z}$ *is symmetric positive semidefinite with* $\ker(\boldsymbol{H_z}) = \mathrm{span}\{\boldsymbol{1}\}$ *and* $\mathrm{rank}(\boldsymbol{H_z}) = V - 1$. *Moreover, for the residual* $\boldsymbol{g}$ *we have* $\boldsymbol{1}^\top\boldsymbol{g} = 0$, *hence* $\boldsymbol{g} \in \boldsymbol{1}^\perp = \mathrm{range}(\boldsymbol{H_z})$; *in particular, given any eigenbasis of* $\boldsymbol{H_z}$ *restricted to* $\boldsymbol{1}^\perp$, $\boldsymbol{g}$ *admits a unique coordinate representation in that basis.*

**Corollary 3.4** (Modal dynamics in the eigenbasis of $\boldsymbol{H_z}^t$). *Under the same assumptions as Theorem 3.2. For each* $t$, *let the spectral decomposition of the symmetric positive semidefinite matrix* $\boldsymbol{H_z}^t$ *be*

$$\boldsymbol{H_z}^t = \sum_{k=1}^{V-1} \lambda_k^t\,\boldsymbol{v}_k^t(\boldsymbol{v}_k^t)^\top,$$

*where* $\lambda_k^t > 0$, $(\boldsymbol{v}_k^t)^\top\boldsymbol{v}_\ell^t = \delta_{k\ell}$ *are the non-zero eigenvalues and eigenvectors. Define the* modal coefficients *of the residual* $\boldsymbol{g}^t = \boldsymbol{p}^t - \boldsymbol{y}$ *by*

$$e_k^t := (\boldsymbol{v}_k^t)^\top\boldsymbol{g}^t, \quad e_k^{t+1} := (\boldsymbol{v}_k^t)^\top\boldsymbol{g}^{t+1}, \qquad k = 1, \ldots, V-1. \tag{4}$$

*Then there exists a constant* $C > 0$ *such that for all nonzero modes* $k \geq 1$,

$$e_k^{t+1} = \Big(1 - \eta\,\mu\,\big[\lambda_k^t + \underbrace{\tilde{\rho}^t(\lambda_k^t)^2}_{\text{SAM's correction}}\big]\Big)e_k^t + r_k^t, \qquad |r_k^t| \leq C\,\eta^2. \tag{5}$$

Proofs are deferred to Appendix B. The corollary diagonalizes the vector dynamics into coordinate-wise scalars in the eigenbasis of $\boldsymbol{H_z}$, making SAM's effect transparent. We now characterize the additional SAM correction in two regimes.

- **Case 1: Positive** $\eta$ (corresponding to the $\boldsymbol{y}^+$ objective in DPO). In this case, GD induces a stronger contraction of the residual $\boldsymbol{g}$ along high-curvature directions, i.e., those associated with large eigenvalues of $\boldsymbol{H_z}$. The additional correction term introduced by SAM has the same sign as that of GD, thereby amplifying this contraction.
- **Case 2: Negative** $\eta$ (corresponding to the $\boldsymbol{y}^-$ objective in DPO). Here, GD causes the residual $\boldsymbol{g}$ to expand more rapidly along high-curvature directions. Moreover, standard SAM with positive $\rho$ exacerbates this phenomenon, leading to even faster expansion compared to GD. By contrast, choosing a negative $\rho$ counteracts this expansion.

Next, we extend our theoretical framework to the result of Ren & Sutherland (2024), which introduced the squeezing effect. For consistency with their notation, we let $y \in \{y^+, y^-\}$ denote the ground–truth class index (with one–hot label $\boldsymbol{y} = \boldsymbol{e}_y$), and $y^* = \arg\max_{j \neq y} p_j^t$ be the most confident incorrect class.

To compare GD and SAM, we study the *one–step confidence ratio*

$$\alpha_i^{\mathrm{GD}} := \frac{p_i^{t+1}(\mathrm{GD})}{p_i^t}, \qquad \alpha_i^{\mathrm{SAM}} := \frac{p_i^{t+1}(\mathrm{SAM})}{p_i^t},$$

which measures the multiplicative change of the class probability after one update.

**Lemma 3.5** (One–step confidence ratios under GD, Lemma 1 of Ren & Sutherland (2024))**.** *Consider the negative learning–rate case $\eta < 0$, for which the ground–truth label is $y = y^-$. Then*

$$\alpha_{y^*}^{\mathrm{GD}} > 1, \qquad \alpha_{y^-}^{\mathrm{GD}} < 1.$$

Lemma 3.5 formalizes the squeezing effect under GD with a negative learning rate: the probability of the most confident incorrect class increases, while that of the ground–truth class decreases. Within our framework, we next analyze the ratio of these two probabilities after a one–step SAM update.

**Corollary 3.6** (One–step confidence ratios under SAM, informal version of Corollary B.7 and Corollary B.10)**.** *Under the same assumptions as Theorem 3.2, set $\eta\rho > 0$ and $|\rho| = \kappa\sqrt{|\eta|}$. Then, for sufficiently small step size $|\eta|$, the following inequality holds:*

$$\alpha_{y^*}^{\mathrm{SAM}} \ \leq \ \alpha_{y^*}^{\mathrm{GD}}, \tag{6}$$

*and under additional technical conditions, the following inequality holds:*

$$\alpha_y^{\mathrm{SAM}} \ \geq \ \alpha_y^{\mathrm{GD}}. \tag{7}$$

*Here $y \in \{y^+, y^-\}$ denotes the ground–truth label corresponding to the positive or negative learning rate, respectively. Moreover, the inequalities are strict whenever $p_{y^*}^t \in (0, 1)$ and $\tilde{\rho}^t \neq 0$.*

The proof is deferred to Appendix B. Corollary 3.6 and Lemma 3.5 together imply that, when $\eta < 0$, using SAM with a negative $\rho < 0$ moderates the growth of the most confident incorrect class and slows the decay of the ground-truth class, thereby preventing excessive expansion and premature collapse. Thus, for negative $\eta$, choosing a *negative* $\rho$ effectively alleviates the squeezing effect.

We empirically validate this prediction using a 1000-dimensional toy example with three classes. We first train for 10 epochs using class 0 as the label to mimic SFT, and then switch to class 1 while continuing training with a negative learning rate. As shown in Figure 1, this setup reproduces the squeezing effect observed in prior work (Ren & Sutherland, 2024): both modal coefficients expand rapidly, the probabilities of class 1 and class 2 decrease, and only the probability of class 0 increases. Moreover, SAM with positive $\rho$ exacerbates this behavior, whereas SAM with negative $\rho$ suppresses it, exactly as predicted by our theory.

Additionally, Corollary 3.6 indicates that for $\eta > 0$, using SAM with $\rho > 0$ also has a beneficial effect on alleviating the squeezing effect: the contraction of $y^*$ is accelerated, while the growth of the ground-truth $y^+$ is enhanced. Taken together, these results suggest a simple rule: choosing $\rho$ with the *same sign* as the learning rate alleviates the squeezing effect.

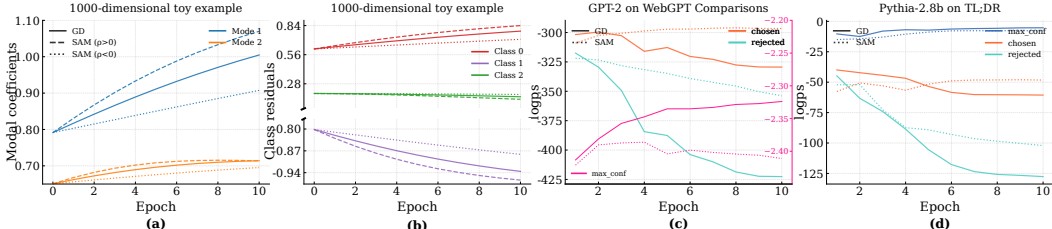

Figure 1: Training dynamics under different settings. (a–b) 1000-dimensional toy example with three classes, trained with a negative learning rate under GD, SAM ($\rho > 0$), and SAM ($\rho < 0$). Panel (a) shows the modal coefficients, and panel (b) shows the class residuals. (c) Real-data experiment on WebGPT Comparisons with GPT-2, comparing GD and SAM: the panel reports the log-probabilities of the chosen responses, the rejected responses, and `max_conf`, which denotes the model's most confident response. (d) Real-data experiment on the TL;DR dataset with Pythia-2.8B, showing the same three curves (chosen, rejected, and `max_conf`).

To verify this rule in real models, we track the probability dynamics of chosen, rejected, and most confident responses when fine-tuning GPT-2 (Radford et al., 2019) on the WebGPT Comparisons dataset (Nakano et al., 2022) (Figure 1c) and Pythia-2.8B (Biderman et al., 2023) on the TL;DR dataset (Stiennon et al., 2020) (Figure 1d). In both cases, SAM increases the probability of chosen responses, slows the decrease of rejected responses, and prevents the most confident responses from growing, consistent with our theoretical predictions.

## 3.3 FROM THEORY TO PRACTICE

In practice, an important challenge in applying SAM to DPO is that it requires an additional forward and backward pass, thereby nearly doubling the computational cost. However, our dynamical analysis shows that curvature regularization can still be achieved even when the perturbation is applied solely in the logit space (with an appropriate choice of the sign of $\rho$), which also alleviates the squeezing effect. Motivated by this observation, we suggest using a computationally efficient SAM variant that perturbs only in the last layer, called *logits-SAM*, to improve the effectiveness and robustness of DPO. Its objective can be formulated as follows:

$$\mathcal{L}_{\text{DPO}}^{\text{logits-SAM}}(\boldsymbol{W}, \boldsymbol{\theta}; \boldsymbol{x}, \boldsymbol{y}^+, \boldsymbol{y}^-) = \mathcal{L}_{\text{DPO}}\left(\boldsymbol{W} + \rho \frac{\nabla_{\boldsymbol{W}} \mathcal{L}_{\text{DPO}}(\boldsymbol{W}, \boldsymbol{\theta}; \boldsymbol{x}, \boldsymbol{y}^+, \boldsymbol{y}^-)}{\left\|\nabla_{\boldsymbol{W}} \mathcal{L}_{\text{DPO}}(\boldsymbol{W}, \boldsymbol{\theta}; \boldsymbol{x}, \boldsymbol{y}^+, \boldsymbol{y}^-)\right\|}, \boldsymbol{\theta}; \boldsymbol{x}, \boldsymbol{y}^+, \boldsymbol{y}^-\right).$$

where $\boldsymbol{W}$ denotes the parameters in the output layer, and $\boldsymbol{\theta}$ denotes the parameters except $\boldsymbol{W}$.

**Implementation.** Unlike our theoretical setting, common DPO implementations[1][2] typically encode the $\boldsymbol{y}^-$ objective as negative while using a single positive learning rate, rather than assigning positive and negative rates to $\boldsymbol{y}^+$ and $\boldsymbol{y}^-$, respectively. Accordingly, we adopt this convention in our logits-SAM implementation. Our dynamical analysis further indicates that $\rho$ should share the sign of the learning rate; hence we consistently use a positive $\rho$. In Appendix D, we provide the full update formulas and a derivation establishing the equivalence between the theoretical and practical settings, summarized in Table 5.

*Remark.* This choice does not render our analysis of the negative learning rate redundant. For first-order methods such as GD, using a negative objective with a positive learning rate is equivalent to using a positive objective with a negative learning rate. Therefore, our analysis applies fully to the case of negative objectives.

The implementation pseudocode can be found in Algorithm 1 of Appendix C. We compute the perturbation manually using the hidden states from the penultimate layer and the parameters of the final layer, requiring only a single full forward–backward pass instead of the two full passes required in standard SAM. Since the parameters of the final layer typically constitute only a small fraction of all trainable parameters (e.g., 4.64% in Pythia-2.8B and 1.81% in Mistral-7B), the additional

---

[1]`https://github.com/eric-mitchell/direct-preference-optimization`
[2]`https://github.com/huggingface/trl`

training overhead introduced by logits-SAM is negligible. A detailed comparison of wall-clock time and peak memory usage is provided in Section 4.3.

# 4 EXPERIMENTS

## 4.1 EXPERIMENTAL SETUP

**Datasets.** We conduct DPO training on three widely used datasets to evaluate our algorithm: Anthropic-HH (Bai et al., 2022), the Reddit TL;DR summarization dataset (Stiennon et al., 2020), and the UltraFeedback Binarized dataset (Cui et al., 2023).

**Models.** Following common practice, we adopt SFT models as our base models. We use Pythia-2.8B (Biderman et al., 2023) for experiments on Anthropic-HH and Reddit TL;DR, and Mistral-7B-v0.1 (Jiang et al., 2023) for UltraFeedback. For Pythia-2.8B, we initialize from the Hugging Face open-source checkpoint[3], which was SFT for one epoch on Anthropic-HH. For the TL;DR experiments, we use the checkpoint[4], which was SFT for one epoch on Reddit TL;DR. For Mistral-7B-v0.1, we use the Alignment Handbook (Tunstall et al., 2023a) checkpoint Zephyr-7b[5] (Tunstall et al., 2023b), which was SFT for one epoch on UltraChat-200k (Ding et al., 2023).

**Evaluation.** For Pythia-2.8B, we evaluate model performance on Anthropic-HH and Reddit TL;DR by measuring win rates (WR) against both the SFT baseline and the human-preferred responses, using GPT-5-mini (version 2025-08-07) as the automatic judge. Following the DPO paper, we set the decoding temperature to 1 for HH and 0 for TL;DR. For Mistral-7B-v0.1, we conduct evaluation on three popular open-ended instruction-following benchmarks: AlpacaEval 2 (Dubois et al., 2024), Arena-Hard v0.1 (Li et al., 2024), and MT-Bench (Zheng et al., 2023). Details of each benchmark can be found in Appendix E. We adopt the default generation parameters provided by each benchmark. Specifically, we report both length-controlled win rates (LC) and raw WR for AlpacaEval 2, model WR for Arena-Hard v0.1, and averaged judge scores (1–10) for MT-Bench, all following the standard evaluation protocols.

**Baselines.** We apply logits-SAM to DPO and two SOTA variants, SLiC-HF (Zhao et al., 2023) and CPO (Xu et al., 2024). We use AdamW optimizer (Loshchilov & Hutter, 2019) in all experiments. For Pythia-2.8B, we set batch size 64 and learning rate $1 \times 10^{-6}$, following the DPO paper; for Mistral-7B, we use batch size 128 and learning rate $5 \times 10^{-7}$, following the Alignment Handbook's recommended settings.

**Hyperparameters.** For DPO, we adopt the recommended $\beta$ values from the DPO paper and the Alignment Handbook, which are widely used and well tuned. For SLiC-HF and CPO, we select hyperparameters following the tuning protocol from Meng et al. (2024b). For logits-SAM, we keep all hyperparameters identical to each corresponding baseline to ensure fairness; the only additional hyperparameter is $\rho$, which we tune over $\{1 \times 10^{-5}, 1 \times 10^{-4}, 1 \times 10^{-3}\}$. Full hyperparameter settings are provided in Table 6 and Table 7 of Appendix E.

## 4.2 EXPERIMENTAL RESULTS

**Performance of summarization and dialogue generation tasks.** We present the results in Table 1. We find that logits-SAM consistently improves performance across both HH and TL;DR datasets. All three baselines (DPO, SLiC-HF, and CPO) achieve higher win rates against both SFT and chosen responses when augmented with logits-SAM. Notably, SLiC-HF shows the largest gains on HH (+6.60 pp vs SFT, +7.49 pp vs chosen), while CPO achieves strong improvements on TL;DR (+2.30 pp vs SFT, +6.03 pp vs chosen), demonstrating that logits-SAM provides stable and generalizable benefits across different optimization methods.

---

[3] https://huggingface.co/lomahony/eleuther-pythia2.8b-hh-sft
[4] https://huggingface.co/trl-lib/pythia-2.8b-deduped-tldr-sft
[5] https://huggingface.co/alignment-handbook/zephyr-7b-sft-full

Table 1: Evaluation results (WR %) on HH and TL;DR datasets using Pythia-2.8B. The judge is GPT-5-mini. The highest value within each method group (baseline vs. logits-SAM) is **bolded**.

| Method | HH | | TL;DR | |
| --- | --- | --- | --- | --- |
| | vs SFT | vs chosen | vs SFT | vs chosen |
| DPO | 70.52 | 56.35 | 84.21 | 34.78 |
| DPO+logits-SAM | **72.28** | **60.51** | **89.58** | **36.57** |
| SLiC-HF | 65.27 | 54.72 | 91.88 | 31.36 |
| SLiC-HF+logits-SAM | **71.87** | **62.21** | **94.40** | **32.80** |
| CPO | 66.60 | 58.19 | 90.99 | 39.38 |
| CPO+logits-SAM | **70.24** | **59.90** | **93.29** | **45.41** |

Table 2: Evaluation results on AlpacaEval 2 (LC and WR), Arena-Hard v0.1 (WR), and MT-Bench using Mistral-7B-v0.1. Judges are GPT-4 Turbo for AlpacaEval 2, and GPT-4.1 for Arena-Hard v0.1 and MT-Bench. The highest value within each method group (baseline vs. logits-SAM) is **bolded**.

| Method | AlpacaEval 2 | | Arena-Hard v0.1 | MT-Bench |
| --- | --- | --- | --- | --- |
| | LC (%) | WR (%) | WR (%) | (score) |
| DPO | 13.08 | 10.96 | 19.0 | 5.49 |
| DPO+logits-SAM | **13.90** | **11.62** | **23.1** | **5.79** |
| SLiC-HF | 8.92 | 8.97 | 19.1 | 5.05 |
| SLiC-HF+logits-SAM | **10.63** | **9.23** | **21.1** | **5.22** |
| CPO | 8.97 | 8.13 | 19.2 | 5.22 |
| CPO+logits-SAM | **13.32** | **11.78** | **21.4** | **5.49** |

**Performance on open-ended instruction-following benchmarks.** We present the results in Table 2. The results demonstrate that combining logits-SAM with different DPO variants consistently yields performance gains across all benchmarks. On open-ended instruction-following evaluations, logits-SAM improves both length-controlled and original win rates on AlpacaEval 2 (e.g., with CPO: +4.35 pp LC, +3.65 pp WR), increases head-to-head win rate on Arena-Hard v0.1 (e.g., with DPO: +4.1 pp WR), and provides steady gains on MT-Bench (e.g., DPO: +0.30, SLiC-HF: +0.17, CPO: +0.27). These findings indicate that logits-SAM is a generally effective and robust enhancement across diverse evaluation settings.

## 4.3 ADDITIONAL ANALYSIS

**Computational overhead.** Compared to vanilla SAM, logits-SAM minimizes additional computational overhead. We report wall-clock training time and peak memory on Pythia-2.8B trained on the Reddit TL;DR dataset (Figure 2), using data-parallel training (DDP) across two NVIDIA A100 GPUs with a per-device batch size of 4. The results show that logits-SAM adds only $\sim 2$–$3\%$ extra time, with negligible peak-memory overhead. By contrast, vanilla SAM is practically infeasible for Pythia-2.8B on A100s with DDP: it nearly doubles the step time (due to an extra full forward–backward pass) and requires

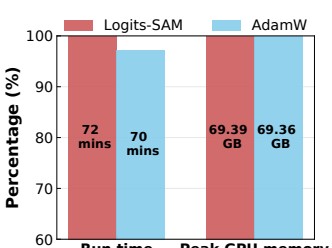

Figure 2: Efficiency comparison.

a perturbation buffer comparable to the model size (for billion-parameter models, this entails more than 10 GB of additional GPU memory), which leads to out-of-memory even with batch size 1. These observations highlight the clear computational cost advantage of logits-SAM.

**Sensitivity analysis.** We present a sensitivity analysis of the additional hyperparameter $\rho$ for logits-SAM in Table 3. The results indicate that, within a reasonable range of $\rho$, performance is typically improved, whereas further enlarging $\rho$ leads to a marked degradation. Notably, unlike original SAM, logits-SAM perturbs only the output layer, so the appropriate scale of $\rho$ is much smaller than the range (0.01–0.5) recommended in the SAM paper. We recommend starting the search for logits-SAM's $\rho$ at $10^{-5}$ or $10^{-4}$ and, if resources permit, performing a finer sweep in this neighborhood.

Table 3: Performance on HH and TL;DR datasets under different $\rho$ values. Each entry reports win rate vs SFT (left) and vs chosen (right).

| Dataset | $\rho = 0$ (AdamW) | $\rho = 10^{-5}$ | $\rho = 10^{-4}$ | $\rho = 10^{-3}$ | $\rho = 10^{-2}$ |
|---------|-------------------|------------------|------------------|------------------|------------------|
| HH | 70.52 / 56.35 | 69.47 / 58.27 | 72.28 / 60.51 | 68.49 / 59.52 | 65.49 / 56.31 |
| TL;DR | 84.21 / 34.78 | 87.79 / 33.97 | 89.58 / 36.57 | 84.25 / 29.93 | 81.56 / 29.31 |

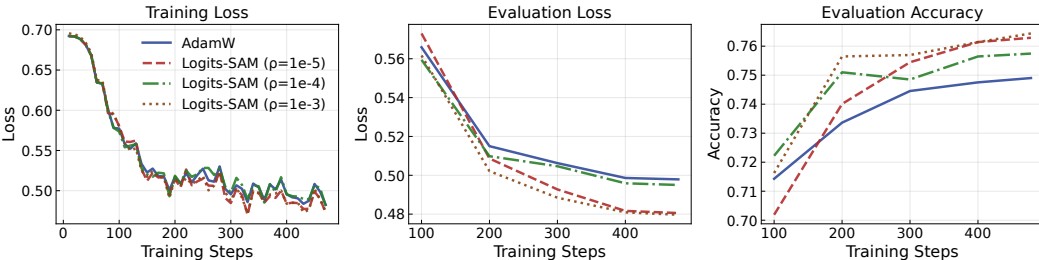

Figure 3: Learning dynamics of Mistral-7B on UltraFeedback. We compare AdamW and logits-SAM in terms of training loss, evaluation loss, and evaluation accuracy, and report curves for logits-SAM under different values of $\rho$.

**Learning dynamics.** In Figure 3, we compare the learning dynamics of AdamW and logits-SAM when training Mistral-7B on the UltraFeedback dataset. The figure reports training loss, evaluation loss, and evaluation accuracy across training steps, and includes curves for logits-SAM under multiple choices of $\rho$ ($1 \times 10^{-5}, 1 \times 10^{-4}, 1 \times 10^{-3}$). We observe that for all three values of $\rho$, logits-SAM achieves training loss that is fairly close to that of AdamW, yet consistently attains lower evaluation loss and higher evaluation accuracy. This indicates that logits-SAM provides better generalization than AdamW, and the fact that a range of $\rho$ values yields consistent improvements suggests that its benefits are robust to the choice of this hyperparameter.

**Sharpness.** To further probe the reasons underlying the generalization gains of logits-SAM, we measure the traces of the parameter Hessian and the logit Hessian at the final checkpoint of Mistral-7B. For AdamW, the traces are $1.337 \times 10^4$ / $2.732 \times 10^2$ (parameter / logit Hessian), while for logits-SAM they are reduced to $1.186 \times 10^4$ / $2.586 \times 10^2$. This reduction indicates that logits-SAM converges to a flatter solution, which is widely believed to be beneficial for generalization.

**Extension to AI safety and on-policy setting.** Razin et al. (2024) refer to the squeezing effect as *likelihood displacement* and find that, in AI safety scenarios, it can reduce the model's harmful-response refusal rate, which leads to severe safety concerns. We evaluate the performance of logits-SAM in the same on-policy setting and AI safety scenario as in their work. The reference model is the instruction-tuned Gemma-2B-IT (Team et al., 2024), and the evaluation is conducted using SorryBench (Xie et al., 2024). We train for one epoch with a learning rate of $1 \times 10^{-6}$ and a batch size of 16. We compare the performance of the reference model, DPO, DPO with logits-SAM, CHES (Razin et al., 2024), and CHES with logits-SAM. For CHES, we filter 50% of samples using the CHES score. Performance is measured by the harmful-response refusal rate and is reported in Table 4. The results show that logits-SAM significantly improves performance in this setting. In particular, DPO with logits-SAM avoids the degradation in refusal rate and performs better than the reference model. Combining logits-SAM with the CHES method of Razin et al. (2024) further increases the refusal rate, with an absolute improvement of approximately 9% on both the training and test sets. These findings indicate that logits-SAM can be effectively transferred to other settings and tasks.

Table 4: Train and test refusal rates for different methods on SorryBench (higher is better).

|  | Ref model | DPO | DPO+logits-SAM | CHES | CHES+logits-SAM |
|---|-----------|-----|----------------|------|-----------------|
| Train Refusal | 0.8054 | 0.7703 | **0.8135** | 0.8459 | **0.9324** |
| Test Refusal | 0.7231 | 0.7077 | **0.7538** | 0.7846 | **0.8769** |

## 5 RELATED WORK

**Reinforcement learning from human feedback.** RLHF has emerged as the de facto post-training recipe for aligning large language models (Christiano et al., 2017; Ziegler et al., 2019; Ouyang et al., 2022; Bai et al., 2022), typically combining supervised fine-tuning (Zhou et al., 2023; Taori et al., 2023; Conover et al., 2023; Wang et al., 2023b), reward modeling (Gao et al., 2023; Luo et al., 2023; Lambert et al., 2024), and policy optimization (Schulman et al., 2017; Anthony et al., 2017). To reduce the complexity and instability of online preference optimization, offline methods such as SLiC-HF (Zhao et al., 2023) and RRHF (Yuan et al., 2023) learn policies from comparisons using closed-form objectives. DPO (Rafailov et al., 2024b) is a central example that maximizes the log-probability margin between preferred and rejected responses relative to a reference policy. Thanks to its simplicity and training stability, DPO has rapidly gained popularity, spurring a line of variants aimed at improving performance. For example, Azar et al. (2024) propose IPO, a more theoretically grounded variant; CPO (Xu et al., 2024) approximates the reference policy as uniform to eliminate the reference term; f-DPO (Wang et al., 2023a) generalizes DPO via a family of $f$-divergences; SimPO (Meng et al., 2024a) uses length-normalized scores that better reflect generation-time preferences; and Cal-DPO (Xiao et al., 2024) aligns the implicit reward scale with likelihoods.

**Squeezing effect (likelihood displacement).** The squeezing effect (Ren & Sutherland, 2024), also known as likelihood displacement (Razin et al., 2024), refers to the recently identified phenomenon in which the probability of the ground-truth label is unintentionally reduced during DPO training. This effect has been widely observed and can lead to performance degradation, reduced safety, and even alignment failure (Pal et al., 2024; Yuan et al., 2024; Rafailov et al., 2024a; Tajwar et al., 2024; Pang et al., 2024). Several studies have attempted to mitigate this issue. Asadi et al. (2025) constrain the shift of probability mass between preferred and rejected responses in the reference and target policies. Liu et al. (2025) introduce a KL-divergence-based policy drift constraint to dynamically regularize policy updates. Razin et al. (2024) strengthen safety alignment by filtering samples that are likely to induce likelihood displacement based on the CHES score between token embeddings. Unlike existing approaches, which either focus on designing alternative objective functions or filter the training data, our method takes a pure optimization-based perspective. It is therefore conceptually orthogonal to these techniques and can be used in combination with them.

**Sharpness-aware minimization.** A widely held belief in the deep learning community is that flatter solutions typically generalize better (Hochreiter & Schmidhuber, 1997; Keskar et al., 2016; Dinh et al., 2017; Jiang et al., 2019; Xie et al., 2020; Liu et al., 2023). Motivated by this view, SAM (Foret et al., 2021) is a bilevel optimization method that explicitly seeks flatter minima, and it has gained popularity for delivering consistent improvements across a wide range of supervised learning tasks (Foret et al., 2021; Kwon et al., 2021; Kaddour et al., 2022; Liu et al., 2022; Kim et al., 2022; Li & Giannakis, 2023; Nguyen et al., 2023a;b; Truong et al., 2024; Luo et al., 2025b; Phan et al., 2025; Luo et al., 2025a). Most relevant to our work are its recent applications in LLMs. Singh et al. (2025) propose Functional-SAM for LLM pretraining and demonstrate strong performance, while Lee & Yoon (2025) apply SAM to Proximal Policy Optimization to improve robustness in both the reward and action spaces. Logits-SAM is a byproduct mentioned in recent studies, yet it is often overlooked. Baek et al. (2024) analyze the effect of label noise on SAM in linear regression and argue that Jacobian-SAM, the counterpart of logits-SAM, plays the dominant role. Similarly, Singh et al. (2025) identify Jacobian-SAM, also referred to as Functional-SAM, as more important and show that it can effectively improve the generalization performance of LLM pretraining.

## 6 CONCLUSION

We analyzed the squeezing effect in DPO via coordinate-wise dynamics in parameter and logit spaces. Our framework shows that GD with negative $\eta$ drives residuals to expand along high-curvature directions, and that SAM suppresses this behavior via curvature regularization; in particular, negative $\eta$ calls for negative $\rho$. Motivated by this, we adopt *logits-SAM*, which perturbs only the output layer and adds negligible overhead, and demonstrate consistent gains in effectiveness and robustness across models and datasets. We expect these insights to inform curvature-aware preference optimization going forward.

## ACKNOWLEDGEMENTS

Trung Le, Mehrtash Harandi, and Dinh Phung were supported by the ARC Discovery Project grant DP250100262. Trung Le and Mehrtash Harandi were also supported by the Air Force Office of Scientific Research under award number FA9550-23-S-0001.

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

## A  ADDITIONAL RELATED WORK

**Kernel and fixed feature regime of LLMs.**    In the context of LLMs, there is a growing body of work that investigates model dynamics through the lens of kernels. A pioneering line of work by Malladi et al. (2023) uses Neural Tangent Kernel-based dynamics (Jacot et al., 2018) to accurately characterize the behavior of LLM fine-tuning and achieves performance comparable to fine-tuning through kernel methods, under the fixed feature assumption. Afzal et al. (2024) leverage the spectrum of the NTK to predict the generalization performance of LLM fine-tuning, and Jang et al. (2024) study the training dynamics of low-rank adaptation from an NTK perspective.

## B  FORMAL THEOREMS AND PROOFS

**Notation.**    We use $\|\cdot\|$ to denote the $\ell_2$ norm for vectors and the Frobenius norm for matrices. For linear and multilinear maps, $\|\cdot\|$ denotes the corresponding operator norm induced by these norms.

**Proposition B.1** (Geometry of the logit space and the parameter-logit correspondence)**.** *Let* $\ell :$ $\mathbb{R}^V \to \mathbb{R}$ *be* $C^2$. *Fix an input* $\boldsymbol{x}$ *and a feature map* $\phi(\boldsymbol{x}) \in \mathbb{R}^d$. *For* $\boldsymbol{W} \in \mathbb{R}^{V\times d}$ *set*

$$\boldsymbol{z} = \boldsymbol{W}\,\phi \in \mathbb{R}^V, \qquad F(\boldsymbol{W}) = \ell(\boldsymbol{z}).$$

*Let* $\boldsymbol{H_z} := \nabla_{\boldsymbol{z}}^2 \ell(\boldsymbol{z})$ *and* $\boldsymbol{H_W} := \nabla_{\boldsymbol{W}}^2 F(\boldsymbol{W})$ *be the Hessians. Equip* $\mathbb{R}^{V\times d}$ *with the Frobenius inner product* $\langle \boldsymbol{A}, \boldsymbol{B} \rangle_F = \mathrm{tr}(\boldsymbol{A}^\top \boldsymbol{B})$ *and* $\mathbb{R}^V$ *with the Euclidean inner product. Let*

$$T_\phi : \mathbb{R}^{V\times d} \to \mathbb{R}^V, \qquad T_\phi(\Delta\boldsymbol{W}) = \Delta\boldsymbol{W}\,\phi$$

*be the differential of the map* $\boldsymbol{W} \mapsto \boldsymbol{W}\phi$, *and let* $T_\phi^* : \mathbb{R}^V \to \mathbb{R}^{V\times d}$ *be its adjoint with respect to these inner products, i.e.,* $\langle T_\phi(\Delta\boldsymbol{W}),\, \boldsymbol{v} \rangle = \langle \Delta\boldsymbol{W},\, T_\phi^*(\boldsymbol{v}) \rangle_F$ *for all* $\Delta\boldsymbol{W}, \boldsymbol{v}$. *Then* $T_\phi^*(\boldsymbol{v}) = \boldsymbol{v}\,\phi^\top$. *The following statements hold.*

*(1)  **Pullback identity (operator form).***

$$\boldsymbol{H_W} = T_\phi^*\,\boldsymbol{H_z}\,T_\phi$$

*as linear operators on* $\mathbb{R}^{V\times d}$. *Consequently, if* $\phi \neq \boldsymbol{0}$, *then*

$$\mathrm{rank}(\boldsymbol{H_W}) = \mathrm{rank}(\boldsymbol{H_z}).$$

*In canonical Euclidean coordinates,*

$$\nabla_{\boldsymbol{W}} F(\boldsymbol{W}) = \big(\nabla_{\boldsymbol{z}} \ell(\boldsymbol{z})\big)\,\phi^\top, \qquad \boldsymbol{H_W} = \big(\phi\phi^\top\big) \otimes \boldsymbol{H_z}.$$

*(2)  **Pullback identity (bilinear form).** For every* $\Delta\boldsymbol{W}, \Delta\boldsymbol{W}' \in \mathbb{R}^{V\times d}$, *the bilinear forms*

$$\boldsymbol{H_W}[\Delta\boldsymbol{W},\, \Delta\boldsymbol{W}'] = \boldsymbol{H_z}[T_\phi(\Delta\boldsymbol{W}),\, T_\phi(\Delta\boldsymbol{W}')]$$

*Thus the second–order effect of any parameter perturbation depends only on the induced logits perturbation* $T_\phi(\Delta\boldsymbol{W}) = \Delta\boldsymbol{W}\,\phi$.

*(3)  **Surjectivity, kernel, and quotient-space view.** If* $\phi \neq \boldsymbol{0}$, *then* $T_\phi$ *is surjective. For any* $\Delta\boldsymbol{z} \in \mathbb{R}^V$, *a minimum-Frobenius-norm preimage is*

$$\Delta\boldsymbol{W}_\star = \frac{\Delta\boldsymbol{z}\,\phi^\top}{\|\phi\|^2} \quad with \quad T_\phi(\Delta\boldsymbol{W}_\star) = \Delta\boldsymbol{z}.$$

*The kernel is*

$$\mathrm{ker}(T_\phi) = \{\,\Delta\boldsymbol{W} \in \mathbb{R}^{V\times d} :\ \Delta\boldsymbol{W}\,\phi = \boldsymbol{0}\,\},$$

*of dimension* $V(d-1)$. *Consequently,* $\boldsymbol{H_W}$ *descends to the quotient* $\mathbb{R}^{V\times d}/\mathrm{ker}(T_\phi) \cong \mathbb{R}^V$.

*Proof.*  A direct computation gives

$$\langle T_\phi(\Delta\boldsymbol{W}), \boldsymbol{v} \rangle = \mathrm{tr}\big((\Delta\boldsymbol{W}\phi)^\top \boldsymbol{v}\big) = \mathrm{tr}\big(\Delta\boldsymbol{W}^\top \boldsymbol{v}\,\phi^\top\big) = \langle \Delta\boldsymbol{W},\, \boldsymbol{v}\,\phi^\top \rangle_F,$$

hence

$$T_\phi^*(\boldsymbol{v}) = \boldsymbol{v}\,\phi^\top.$$

**(1) Pullback identity (operator form).** Let $F(\boldsymbol{W}) = \ell(\boldsymbol{W}\phi)$. The first differential of $F$ is
$$\mathrm{d}F[\Delta\boldsymbol{W}] = \big\langle \nabla_{\boldsymbol{z}}\ell(\boldsymbol{z}),\ T_\phi(\Delta\boldsymbol{W}) \big\rangle = \big\langle T_\phi^*\big(\nabla_{\boldsymbol{z}}\ell(\boldsymbol{z})\big),\ \Delta\boldsymbol{W} \big\rangle_F,$$
so
$$\nabla_{\boldsymbol{W}}F(\boldsymbol{W}) = T_\phi^*\big(\nabla_{\boldsymbol{z}}\ell(\boldsymbol{z})\big) = \big(\nabla_{\boldsymbol{z}}\ell(\boldsymbol{z})\big)\phi^\top.$$
Differentiating once more and using $\mathrm{d}(\nabla_{\boldsymbol{z}}\ell)(\boldsymbol{z})[\Delta\boldsymbol{z}] = \boldsymbol{H}_{\boldsymbol{z}}\,\Delta\boldsymbol{z}$ with $\Delta\boldsymbol{z} = T_\phi(\Delta\boldsymbol{W})$ yields, for all $\Delta\boldsymbol{W}, \Delta\boldsymbol{W}'$,
$$\mathrm{d}^2F[\Delta\boldsymbol{W}, \Delta\boldsymbol{W}'] = \big\langle T_\phi(\Delta\boldsymbol{W}),\ \boldsymbol{H}_{\boldsymbol{z}}\,T_\phi(\Delta\boldsymbol{W}') \big\rangle.$$
By the Riesz representation on $(\mathbb{R}^{V\times d}, \langle\cdot,\cdot\rangle_F)$, this means
$$\boldsymbol{H}_{\boldsymbol{W}} = T_\phi^*\,\boldsymbol{H}_{\boldsymbol{z}}\,T_\phi,$$
as linear operators on $\mathbb{R}^{V\times d}$.

Using $T_\phi^*(\boldsymbol{v}) = \boldsymbol{v}\phi^\top$ and $T_\phi(\Delta\boldsymbol{W}) = \Delta\boldsymbol{W}\phi$, we obtain the explicit operator action
$$\boldsymbol{H}_{\boldsymbol{W}}(\Delta\boldsymbol{W}) = T_\phi^*\big(\boldsymbol{H}_{\boldsymbol{z}}\,(\Delta\boldsymbol{W}\phi)\big) = \big(\boldsymbol{H}_{\boldsymbol{z}}\,(\Delta\boldsymbol{W}\phi)\big)\phi^\top = \boldsymbol{H}_{\boldsymbol{z}}\,\Delta\boldsymbol{W}\,(\phi\phi^\top).$$

For the rank statement, assume $\phi \neq \boldsymbol{0}$. Then $T_\phi$ is surjective and $T_\phi^*$ is injective. Hence
$$\mathrm{rank}(\boldsymbol{H}_{\boldsymbol{W}}) = \mathrm{rank}\big(T_\phi^*\,\boldsymbol{H}_{\boldsymbol{z}}\,T_\phi\big) = \mathrm{rank}\big(\boldsymbol{H}_{\boldsymbol{z}}\,T_\phi\big) = \mathrm{rank}(\boldsymbol{H}_{\boldsymbol{z}}),$$
because $\mathrm{range}(T_\phi) = \mathbb{R}^V$.

**Kronecker (matrix) form.** Under the canonical identification $\mathbb{R}^{V\times d} \cong \mathbb{R}^{Vd}$ obtained by stacking columns, denote by $\mathrm{vec}(\cdot)$ the corresponding vectorization. Using the standard identity
$$\mathrm{vec}(\boldsymbol{A}\,\boldsymbol{X}\,\boldsymbol{B}) = (\boldsymbol{B}^\top \otimes \boldsymbol{A})\,\mathrm{vec}(\boldsymbol{X}),$$
we have
$$\mathrm{vec}\big(\boldsymbol{H}_{\boldsymbol{W}}(\Delta\boldsymbol{W})\big) = \mathrm{vec}\big(\boldsymbol{H}_{\boldsymbol{z}}\,\Delta\boldsymbol{W}\,(\phi\phi^\top)\big) = \big((\phi\phi^\top) \otimes \boldsymbol{H}_{\boldsymbol{z}}\big)\,\mathrm{vec}(\Delta\boldsymbol{W}).$$
Hence the matrix representation of $\boldsymbol{H}_{\boldsymbol{W}}$ in these coordinates is
$$\boldsymbol{H}_{\boldsymbol{W}} = (\phi\phi^\top) \otimes \boldsymbol{H}_{\boldsymbol{z}}.$$

**(2) Pullback identity (bilinear form).** Recall that the inner products on $\mathbb{R}^{V\times d}$ and $\mathbb{R}^V$ induce bilinear forms
$$\boldsymbol{H}_{\boldsymbol{W}}[\Delta\boldsymbol{W}, \Delta\boldsymbol{W}'] := \langle \Delta\boldsymbol{W},\ \boldsymbol{H}_{\boldsymbol{W}}(\Delta\boldsymbol{W}')\rangle_F, \qquad \boldsymbol{H}_{\boldsymbol{z}}[\Delta\boldsymbol{z}, \Delta\boldsymbol{z}'] := \langle \Delta\boldsymbol{z},\ \boldsymbol{H}_{\boldsymbol{z}}\Delta\boldsymbol{z}'\rangle.$$
By the operator identity $\boldsymbol{H}_{\boldsymbol{W}} = T_\phi^*\,\boldsymbol{H}_{\boldsymbol{z}}\,T_\phi$, for all $\Delta\boldsymbol{W}, \Delta\boldsymbol{W}'$,
$$\begin{aligned}
\boldsymbol{H}_{\boldsymbol{W}}[\Delta\boldsymbol{W}, \Delta\boldsymbol{W}'] &= \langle \Delta\boldsymbol{W},\ \boldsymbol{H}_{\boldsymbol{W}}(\Delta\boldsymbol{W}')\rangle_F \\
&= \big\langle \Delta\boldsymbol{W},\ T_\phi^*\,\boldsymbol{H}_{\boldsymbol{z}}\,T_\phi(\Delta\boldsymbol{W}')\big\rangle_F \\
&= \big\langle T_\phi(\Delta\boldsymbol{W}),\ \boldsymbol{H}_{\boldsymbol{z}}\,T_\phi(\Delta\boldsymbol{W}')\big\rangle \\
&= \boldsymbol{H}_{\boldsymbol{z}}\big[T_\phi(\Delta\boldsymbol{W}),\ T_\phi(\Delta\boldsymbol{W}')\big].
\end{aligned}$$
Thus the second-order effect of any parameter perturbation depends only on the induced logits perturbation $T_\phi(\Delta\boldsymbol{W}) = \Delta\boldsymbol{W}\phi$.

**(3) Surjectivity, kernel and quotient view.** If $\phi \neq \boldsymbol{0}$, then for any $\Delta\boldsymbol{z} \in \mathbb{R}^V$
$$\Delta\boldsymbol{W}_\star = \frac{\Delta\boldsymbol{z}\,\phi^\top}{\|\phi\|^2} \quad \text{satisfies} \quad T_\phi(\Delta\boldsymbol{W}_\star) = \Delta\boldsymbol{z},$$
so $T_\phi$ is surjective. The same choice minimizes the Frobenius norm among all preimages (row-wise Cauchy–Schwarz). The kernel is $\ker(T_\phi) = \{\Delta\boldsymbol{W} : \Delta\boldsymbol{W}\phi = \boldsymbol{0}\}$, and rank–nullity gives $\dim\ker(T_\phi) = V(d-1)$. Finally, if $\Delta\boldsymbol{W}_1 - \Delta\boldsymbol{W}_2 \in \ker(T_\phi)$, then $T_\phi(\Delta\boldsymbol{W}_1) = T_\phi(\Delta\boldsymbol{W}_2)$ and
$$\begin{aligned}
\boldsymbol{H}_{\boldsymbol{W}}[\Delta\boldsymbol{W}_1, \Delta\boldsymbol{W}_1] = \big\langle T_\phi(\Delta\boldsymbol{W}_1),\ \boldsymbol{H}_{\boldsymbol{z}}\,T_\phi(\Delta\boldsymbol{W}_1)\big\rangle &= \big\langle T_\phi(\Delta\boldsymbol{W}_2),\ \boldsymbol{H}_{\boldsymbol{z}}\,T_\phi(\Delta\boldsymbol{W}_2)\big\rangle \\
&= \boldsymbol{H}_{\boldsymbol{W}}[\Delta\boldsymbol{W}_2, \Delta\boldsymbol{W}_2],
\end{aligned}$$
so the bilinear form descends to the quotient $\mathbb{R}^{V\times d}/\ker(T_\phi) \cong \mathbb{R}^V$.

If $\phi = \boldsymbol{0}$ then $T_\phi \equiv 0$ and $\boldsymbol{H}_{\boldsymbol{W}} \equiv \boldsymbol{0}$, the degenerate case. $\qquad\square$

**Theorem B.2** (Dynamics of SAM). *Fix a sample $\boldsymbol{x}$ and set $\mu = \|\phi\|^2 > 0$. Assume:*

- *For each $\boldsymbol{y}$, the loss $f(\cdot, \boldsymbol{y})$ is $C^3$ in $\boldsymbol{z}$ and there exists a constant $L < \infty$ such that for all $\boldsymbol{z}$,*
$$\|\nabla_{\boldsymbol{z}}^k f(\boldsymbol{z}, \boldsymbol{y})\| \le L, \qquad k = 1, 2, 3.$$

- *The step size $|\eta| \in (0, 1]$ and the SAM radius satisfies $|\rho| \le \kappa \sqrt{|\eta|}$ with a constant $\kappa \ge 0$.*

*Write $F(\boldsymbol{W}) \coloneqq f(\boldsymbol{W}\phi, \boldsymbol{y})$ and $\boldsymbol{z} = \boldsymbol{W}\phi$.*

*Consider* standard SAM:
$$\Delta \boldsymbol{W}_{\mathrm{adv}}^t = \rho \frac{\nabla_{\boldsymbol{W}} F(\boldsymbol{W}^t)}{\|\nabla_{\boldsymbol{W}} F(\boldsymbol{W}^t)\|}, \qquad \widetilde{\boldsymbol{W}}^t = \boldsymbol{W}^t + \Delta \boldsymbol{W}_{\mathrm{adv}}^t, \qquad \boldsymbol{W}^{t+1} = \boldsymbol{W}^t - \eta \nabla_{\boldsymbol{W}} F(\widetilde{\boldsymbol{W}}^t).$$

*We adopt the convention that when $\|\boldsymbol{g}^t\| = 0$, the inner perturbation is set to 0, i.e., $\tilde{\rho}^t = 0$. Otherwise, define $\tilde{\rho}^t \coloneqq \rho \sqrt{\mu}/\|\boldsymbol{g}^t\|$. Then, there exists a constant $C > 0$ (depending only on $L, \mu, \kappa$) such that the following expansions hold with $O(\eta^2)$ remainders:*

*(parameters)* $\quad \boldsymbol{W}^{t+1} = \boldsymbol{W}^t - \eta \Big( \boldsymbol{g}^t \phi^\top + \tilde{\rho}^t \boldsymbol{H}_{\boldsymbol{z}}^t \boldsymbol{g}^t \phi^\top \Big) + \boldsymbol{R}_{\boldsymbol{W}}^t, \qquad \|\boldsymbol{R}_{\boldsymbol{W}}^t\| \le C \eta^2,$

*(logits)* $\quad \boldsymbol{z}^{t+1} = \boldsymbol{z}^t - \eta \mu \Big( \boldsymbol{g}^t + \tilde{\rho}^t \boldsymbol{H}_{\boldsymbol{z}}^t \boldsymbol{g}^t \Big) + \boldsymbol{r}_{\boldsymbol{z}}^t, \qquad \|\boldsymbol{r}_{\boldsymbol{z}}^t\| \le C \eta^2,$

*(logit gradient)* $\quad \boldsymbol{g}^{t+1} = \Big( \boldsymbol{I} - \eta \mu \boldsymbol{H}_{\boldsymbol{z}}^t - \eta \mu \tilde{\rho}^t (\boldsymbol{H}_{\boldsymbol{z}}^t)^2 \Big) \boldsymbol{g}^t + \boldsymbol{r}_{\boldsymbol{g}}^t, \qquad \|\boldsymbol{r}_{\boldsymbol{g}}^t\| \le C \eta^2.$

*In particular, for softmax cross-entropy where $\boldsymbol{g}^t = \boldsymbol{p}^t - \boldsymbol{y}$ and*

*(residual)* $\quad \boldsymbol{p}^{t+1} - \boldsymbol{y} = \Big( \boldsymbol{I} - \eta \mu \boldsymbol{H}_{\boldsymbol{z}}^t - \eta \mu \tilde{\rho}^t (\boldsymbol{H}_{\boldsymbol{z}}^t)^2 \Big)(\boldsymbol{p}^t - \boldsymbol{y}) + \boldsymbol{r}_{\boldsymbol{g}}^t, \quad \|\boldsymbol{r}_{\boldsymbol{g}}^t\| \le C \eta^2.$

*Proof.* By Proposition B.1 (operator form),
$$\nabla_{\boldsymbol{W}} F(\boldsymbol{W}) = \boldsymbol{g} \phi^\top, \qquad \boldsymbol{H}_{\boldsymbol{W}}(\Delta \boldsymbol{W}) = \boldsymbol{H}_{\boldsymbol{z}} \Delta \boldsymbol{W} (\phi\phi^\top),$$
and $T_\phi(\Delta \boldsymbol{W}) = \Delta \boldsymbol{W} \phi$ with $\|T_\phi\| \le \|\phi\| = \sqrt{\mu}$. Moreover, by the multilinear chain rule applied to $F(\boldsymbol{W}) = f(\boldsymbol{W}\phi, \boldsymbol{y})$,
$$\nabla_{\boldsymbol{W}}^3 F(\boldsymbol{W})[\Delta_1, \Delta_2, \Delta_3] = \nabla_{\boldsymbol{z}}^3 f(\boldsymbol{z}, \boldsymbol{y})\big[T_\phi(\Delta_1), T_\phi(\Delta_2), T_\phi(\Delta_3)\big], \tag{8}$$
hence the operator norm satisfies
$$\sup_{\boldsymbol{W}} \|\nabla_{\boldsymbol{W}}^3 F(\boldsymbol{W})\| \le \Big( \sup_{\boldsymbol{z}} \|\nabla_{\boldsymbol{z}}^3 f(\boldsymbol{z}, \boldsymbol{y})\| \Big) \|T_\phi\|^3 \le L \mu^{3/2}. \tag{9}$$

**(1) Parameter update.** If $\|\boldsymbol{g}^t\| > 0$, then $\nabla_{\boldsymbol{W}} F(\boldsymbol{W}^t) = \boldsymbol{g}^t \phi^\top$ and $\|\boldsymbol{g}^t \phi^\top\| = \|\boldsymbol{g}^t\| \|\phi\| = \|\boldsymbol{g}^t\| \sqrt{\mu}$, so
$$\Delta \boldsymbol{W}_{\mathrm{adv}}^t = \rho \frac{\boldsymbol{g}^t \phi^\top}{\|\boldsymbol{g}^t\| \sqrt{\mu}}, \qquad \|\Delta \boldsymbol{W}_{\mathrm{adv}}^t\| = |\rho| \le \kappa \sqrt{|\eta|}.$$
(If $\|\boldsymbol{g}^t\| = 0$, our convention sets $\Delta \boldsymbol{W}_{\mathrm{adv}}^t = \boldsymbol{0}$.) A second-order Taylor expansion of $\nabla_{\boldsymbol{W}} F$ at $\boldsymbol{W}^t$ gives, for some $\theta \in (0, 1)$,
$$\nabla_{\boldsymbol{W}} F(\widetilde{\boldsymbol{W}}^t) = \nabla_{\boldsymbol{W}} F(\boldsymbol{W}^t) + \boldsymbol{H}_{\boldsymbol{W}}^t[\Delta \boldsymbol{W}_{\mathrm{adv}}^t] + \tfrac{1}{2} \nabla_{\boldsymbol{W}}^3 F(\boldsymbol{W}^t + \theta \Delta \boldsymbol{W}_{\mathrm{adv}}^t)[\Delta \boldsymbol{W}_{\mathrm{adv}}^t, \Delta \boldsymbol{W}_{\mathrm{adv}}^t].$$
By equation 9 and $\|\Delta \boldsymbol{W}_{\mathrm{adv}}^t\| \le \kappa \sqrt{|\eta|}$,
$$\Big\| \tfrac{1}{2} \nabla_{\boldsymbol{W}}^3 F(\cdot)[\Delta \boldsymbol{W}_{\mathrm{adv}}^t, \Delta \boldsymbol{W}_{\mathrm{adv}}^t] \Big\| \le \tfrac{1}{2} L \mu^{3/2} \|\Delta \boldsymbol{W}_{\mathrm{adv}}^t\|^2 \le C_0 |\eta|,$$
for a constant $C_0 = C_0(L, \mu, \kappa)$. Using the operator identity from Proposition B.1,
$$\boldsymbol{H}_{\boldsymbol{W}}^t[\Delta \boldsymbol{W}_{\mathrm{adv}}^t] = \boldsymbol{H}_{\boldsymbol{z}}^t \Delta \boldsymbol{W}_{\mathrm{adv}}^t (\phi\phi^\top) = \frac{\rho \sqrt{\mu}}{\|\boldsymbol{g}^t\|} \boldsymbol{H}_{\boldsymbol{z}}^t \boldsymbol{g}^t \phi^\top = \tilde{\rho}^t \boldsymbol{H}_{\boldsymbol{z}}^t \boldsymbol{g}^t \phi^\top.$$
Therefore
$$\boldsymbol{W}^{t+1} = \boldsymbol{W}^t - \eta \Big( \boldsymbol{g}^t \phi^\top + \tilde{\rho}^t \boldsymbol{H}_{\boldsymbol{z}}^t \boldsymbol{g}^t \phi^\top \Big) - \eta \boldsymbol{R}_\nabla^t,$$
where $\|\boldsymbol{R}_\nabla^t\| \le C_0 |\eta|$. Setting $\boldsymbol{R}_{\boldsymbol{W}}^t \coloneqq -\eta \boldsymbol{R}_\nabla^t$ yields $\|\boldsymbol{R}_{\boldsymbol{W}}^t\| \le C \eta^2$ with $C = C(L, \mu, \kappa)$, proving the parameter expansion.

**(2) Logit update.** Right-multiplying by $\phi$ and using $\mu = \|\phi\|^2$,

$$z^{t+1} - z^t = (W^{t+1} - W^t)\phi = -\eta\,\mu\Big(g^t + \tilde{\rho}^{\,t}\,H_z^t g^t\Big) + r_z^t,$$

with $\|r_z^t\| \le \|R_W^t\|\,\|\phi\| \le C\,\eta^2$ (absorbing $\sqrt{\mu}$ into $C$). This proves the logits expansion.

**(3) logit gradient update.** Since $g = \nabla_z f(z, y)$, a first-order Taylor expansion at $z^t$ gives

$$g^{t+1} = g^t + H_z^t(z^{t+1} - z^t) + \tfrac{1}{2}\nabla_z^3 f(z^t + \xi^t, y)\big[\Delta z^t, \Delta z^t\big], \quad \Delta z^t = z^{t+1} - z^t.$$

By assumption $\|\nabla_z^3 f\| \le L$ and $\|\Delta z^t\| = O(\eta)$ from the previous step, hence the remainder has norm $\le C_1\,\eta^2$. Substituting the logits expansion from step (ii) yields

$$g^{t+1} = \Big(I - \eta\,\mu\,H_z^t - \eta\,\mu\,\tilde{\rho}^{\,t}\,(H_z^t)^2\Big)g^t + r_g^t, \qquad \|r_g^t\| \le C\,\eta^2,$$

after absorbing constants into $C$. This proves the logit gradient statement.

Combining (i)–(iii) completes the proof, with a constant $C$ depending only on $(L, \mu, \kappa)$, and the bounds holding for all $|\eta| \in (0, 1]$ and $|\rho| \le \kappa\sqrt{|\eta|}$.

For softmax cross-entropy,

$$\nabla_z f(z, y) = p(z) - y, \qquad H_z(z) = \nabla_z^2 f(z, y) = \mathrm{Diag}(p(z)) - p(z)p(z)^\top.$$

Since $p(z) \in \Delta^{V-1} \subset [0, 1]^V$ for all $z$, every entry of the third derivative tensor $\nabla_z^3 f(z, y)$ is a bounded polynomial in $p(z)$ (hence in $[0, 1]$). Therefore there exists a finite constant $L_{\mathrm{sm}}(V)$ depending only on $V$ such that

$$\|\nabla_z^k f(z, y)\| \le L_{\mathrm{sm}}(V), \qquad k = 1, 2, 3.$$

In particular, $f$ is $C^\infty$ and Assumption (1) of the theorem holds with $L = L_{\mathrm{sm}}(V)$. $\qquad \square$

In the remainder of this section, we specialize to the softmax cross-entropy loss.

**Proposition B.3.** $H_z$ *is symmetric positive semidefinite with* $\ker(H_z) = \mathrm{span}\{1\}$ *and* $\mathrm{rank}(H_z) = V - 1$. *Moreover, for the residual $g$ we have* $1^\top g = 0$, *hence* $g \in 1^\perp = \mathrm{range}(H_z)$; *in particular, given any eigenbasis of $H_z$ restricted to $1^\perp$, $g$ admits a unique coordinate representation in that basis.*

*Proof.* Let $p = \mathrm{softmax}(z) \in (0, 1)^V$ so that $1^\top p = 1$, and recall

$$H_z = \mathrm{Diag}(p) - pp^\top.$$

For any $v \in \mathbb{R}^V$,

$$v^\top H_z\, v = \sum_{i=1}^V p_i v_i^2 - \Big(\sum_{i=1}^V p_i v_i\Big)^2 = \mathrm{Var}_p(v) \ge 0,$$

hence $H_z$ is symmetric positive semidefinite. Moreover, $v^\top H_z\, v = 0$ iff $\mathrm{Var}_p(v) = 0$, i.e., $v_i$ is constant across $i$. Since $p_i > 0$ for all $i$, this means $v = c\,1$, thus

$$\ker(H_z) = \mathrm{span}\{1\} \quad \Rightarrow \quad \mathrm{rank}(H_z) = V - \dim\ker(H_z) = V - 1.$$

Then $1^\top g = 1^\top p - 1^\top y = 0$, so $g \in 1^\perp$. For any symmetric matrix, $\mathrm{range}(H_z) = (\ker(H_z))^\perp$; using $\ker(H_z) = \mathrm{span}\{1\}$ yields $1^\perp = \mathrm{range}(H_z)$, hence $g \in \mathrm{range}(H_z)$.

Restrict $H_z$ to the invariant subspace $1^\perp$. Being symmetric, $H_z|_{1^\perp}$ admits an orthonormal eigenbasis $\{v_k\}_{k=1}^{V-1}$ associated with its positive eigenvalues. Since $g \in 1^\perp$, it has the unique expansion $g = \sum_{k=1}^{V-1} e_k\, v_k$, with $e_k = (v_k)^\top g$. $\qquad \square$

**Corollary B.4** (Modal dynamics in the eigenbasis of $H_z^t$)**.** *Under the same assumptions as Theorem B.2. For each $t$, let the spectral decomposition of the symmetric positive–semidefinite matrix $H_z^t$ be*

$$H_z^t = \sum_{k=1}^{V-1} \lambda_k^t\, v_k^t(v_k^t)^\top,$$

where $\lambda_k^t > 0$, $(v_k^t)^\top v_\ell^t = \delta_{k\ell}$ *are the non-zero eigenvalues and eigenvectors. Define the* modal coefficients *of the residual $g^t = p^t - y$ by*

$$e_k^t := (v_k^t)^\top g^t, \qquad k = 1, \ldots, V-1.$$

*Then there exists a constant $C > 0$ such that for all nonzero modes $k \geq 1$,*

$$(v_k^t)^\top g^{t+1} = \left(1 - \eta\,\mu\,[\lambda_k^t + \tilde{\rho}^{\,t}(\lambda_k^t)^2]\right) e_k^t + r_k^t, \qquad |r_k^t| \leq C\,\eta^2. \tag{10}$$

*Proof.* By Theorem B.2 (residual expansion), we have

$$g^{t+1} = \left(I - \eta\,\mu\,H_z^t - \eta\,\mu\,\tilde{\rho}^{\,t}\,(H_z^t)^2\right)g^t + r_g^t, \qquad \|r_g^t\| \leq C\,\eta^2. \tag{11}$$

Fix $t$ and let the eigendecomposition of $H_z^t$ be $H_z^t = \sum_{k=1}^{V-1} \lambda_k^t v_k^t (v_k^t)^\top$ with $\lambda_k^t > 0$ and $\{v_k^t\}_{k=1}^{V-1}$ orthonormal. (The zero mode corresponding to $\lambda = 0$ is orthogonal to $g^t$ in the softmax-CE case and is therefore omitted.)

Project equation 11 onto the eigenvector $v_k^t$:

$$(v_k^t)^\top g^{t+1} = (v_k^t)^\top \left(I - \eta\,\mu\,H_z^t - \eta\,\mu\,\tilde{\rho}^{\,t}\,(H_z^t)^2\right)g^t + (v_k^t)^\top r_g^t.$$

Using the eigen-relations $H_z^t v_k^t = \lambda_k^t v_k^t$ and $(H_z^t)^2 v_k^t = (\lambda_k^t)^2 v_k^t$ and the definition $e_k^t = (v_k^t)^\top g^t$, we obtain

$$(v_k^t)^\top g^{t+1} = \left(1 - \eta\,\mu\,\lambda_k^t - \eta\,\mu\,\tilde{\rho}^{\,t}(\lambda_k^t)^2\right)e_k^t + r_k^t, \qquad r_k^t := (v_k^t)^\top r_g^t.$$

Finally, since $\|v_k^t\| = 1$ we have $|r_k^t| \leq \|r_g^t\| \leq C\,\eta^2$, which is exactly equation 10. This completes the proof. $\qquad\square$

Following Ren & Sutherland (2024), we define the one-step confidence ratio as follows.

**Definition B.5** (One–step confidence ratio). *For each class $i \in [V]$ and update rule $a \in \{\mathrm{GD}, \mathrm{SAM}\}$, define the one–step confidence ratio*

$$\alpha_i^{(a)} := \frac{p_i^{t+1}(a)}{p_i^t}.$$

As a consequence of Theorem B.2, we obtain the following lemma.

**Lemma B.6** (One–step ratio representation and factorization). *Under the assumptions of Theorem B.2, fix an iteration $t$ and set $\eta' = \eta\,\mu$. The ratio $\alpha_i^{(a)}$ admits the representation*

$$\alpha_i^{(a)} = \frac{\sum_{j=1}^V e^{z_j^t}}{\sum_{j=1}^V \beta_j^{(a)}\,e^{z_j^t}},$$

*where*

$$\beta_j^{\mathrm{GD}} = \exp\left\{-\eta'\left[(p_j^t - y_j) - (p_i^t - y_i)\right]\right\},$$

*and the SAM correction appears multiplicatively as*

$$\beta_j^{\mathrm{SAM}} = \underbrace{\exp\{-\eta'(g_j^t - g_i^t)\}}_{\beta_j^{\mathrm{GD}}}\underbrace{\exp\{-\eta'\tilde{\rho}^{\,t}\Delta_{j,i}^t\}}_{\text{curvature factor}}\underbrace{\exp\{r_j^t - r_i^t\}}_{\text{remainder factor}}, \quad \Delta_{j,i}^t := (H_z^t g^t)_j - (H_z^t g^t)_i,$$

*and there exists a constant $C_1 > 0$ that depends only on $(L, \mu, \kappa)$ such that $e^{-2C_1\eta^2} \leq \exp\{r_j^t - r_i^t\} \leq e^{2C_1\eta^2}$ for all $i, j$.*

*Proof.* By Theorem B.2 (logits line),

$$\Delta z := z^{t+1} - z^t = -\eta'\left(g^t + \tilde{\rho}^{\,t} H_z^t g^t\right) + r_z^t, \qquad \|r_z^t\|_\infty \leq \|r_z^t\|_2 \leq C_1\,\eta^2,$$

where $C_1$ depends only on $(L, \mu, \kappa)$ and the hypothesis $|\rho| \leq \kappa\sqrt{|\eta|}$ is in force.

For any increment $\Delta \boldsymbol{z}$,

$$\alpha_i = \frac{p_i(\boldsymbol{z}^t + \Delta \boldsymbol{z})}{p_i(\boldsymbol{z}^t)} = \frac{\sum_j e^{z_j^t}}{\sum_j \exp\{\Delta z_j - \Delta z_i\} e^{z_j^t}} = \frac{\sum_j e^{z_j^t}}{\sum_j \beta_j e^{z_j^t}}.$$

With the above $\Delta \boldsymbol{z}$,

$$\beta_j^{\mathrm{SAM}} = \underbrace{\exp\{-\eta'(g_j^t - g_i^t)\}}_{\beta_j^{\mathrm{GD}}} \underbrace{\exp\{-\eta'\tilde{\rho}^t \Delta_{j,i}^t\}}_{\text{curvature factor}} \underbrace{\exp\{r_j^t - r_i^t\}}_{\text{remainder factor}}, \quad \Delta_{j,i}^t := (\boldsymbol{H}_{\boldsymbol{z}}^t \boldsymbol{g}^t)_j - (\boldsymbol{H}_{\boldsymbol{z}}^t \boldsymbol{g}^t)_i.$$

From $\|\boldsymbol{r}_{\boldsymbol{z}}^t\|_\infty \le C_1 \eta^2$, we have $e^{-2C_1\eta^2} \le \exp\{r_j^t - r_i^t\} \le e^{2C_1\eta^2}$ for all $i, j$. $\qquad \square$

We begin by proving the first part of Corollary 3.6, which relies on the same assumptions as Theorem B.2, except that we set $\rho = \kappa\sqrt{\eta}$ instead of requiring $\rho \le \kappa\sqrt{\eta}$.

**Corollary B.7** (One–step confidence ratio of $y^*$ under SAM vs. GD). *Under the assumptions of Theorem B.2, fix an iteration $t$ and set $\eta' = \eta\mu$ and $|\rho| = \kappa\sqrt{|\eta|}$. Let $y$ be the ground–truth label and $y^* = \arg\max_{j\neq y} p_j^t$ the most confident incorrect class. Assume the sign condition $\eta'\tilde{\rho}^t > 0$. Then there exists $\eta_0 = \eta_0(\boldsymbol{p}^t, \boldsymbol{H}_{\boldsymbol{z}}^t, \|\boldsymbol{g}^t\|, \mu, \kappa, L) > 0$ such that, for all step sizes $0 < |\eta| \le \eta_0$, the following one–step inequalities hold:*

$$\alpha_{y^*}^{\mathrm{SAM}} \le \alpha_{y^*}^{\mathrm{GD}}.$$

*Moreover, the inequality is strict whenever $p_{y^*}^t \in (0, 1)$ and $\tilde{\rho}^t \neq 0$. In particular, when $\tilde{\rho}^t = 0$ (no SAM), equality holds.*

*Proof.* Define

$$D_i^{\mathrm{GD}} := \sum_j \beta_j^{\mathrm{GD}} e^{z_j^t}, \qquad \widetilde{D}_i := \sum_j \beta_j^{\mathrm{GD}} e^{z_j^t} \exp\{-\eta'\tilde{\rho}^t \Delta_{j,i}^t\}, \qquad D_i^{\mathrm{SAM}} := \sum_j \beta_j^{\mathrm{SAM}} e^{z_j^t},$$

with

$$\beta_j^{\mathrm{SAM}} = \beta_j^{\mathrm{GD}} \exp\{-\eta'\tilde{\rho}^t \Delta_{j,i}^t\} \exp\{r_j^t - r_i^t\}, \qquad \Delta_{j,i}^t := (\boldsymbol{H}_{\boldsymbol{z}}^t \boldsymbol{g}^t)_j - (\boldsymbol{H}_{\boldsymbol{z}}^t \boldsymbol{g}^t)_i.$$

By the remainder bounds of Lemma B.6, $e^{-2C_1\eta^2} \widetilde{D}_i \le D_i^{\mathrm{SAM}} \le e^{2C_1\eta^2} \widetilde{D}_i$.

With $\boldsymbol{H}_{\boldsymbol{z}}^t = \mathrm{diag}(\boldsymbol{p}^t) - \boldsymbol{p}^t(\boldsymbol{p}^t)^\top$ and $\boldsymbol{g}^t = \boldsymbol{p}^t - \boldsymbol{e}_y$,

$$(\boldsymbol{H}_{\boldsymbol{z}}^t \boldsymbol{g}^t)_i = p_i^t(p_i^t - y_i - C^t), \qquad C^t := \sum_k (p_k^t)^2 - p_y^t.$$

Then $C^t \le p_{y^*}^t$ and one checks: for $i = y^*$, $\Delta_{j,y^*}^t \le 0$ for all $j$, and $\Delta_{j,y^*}^t < 0$ for some $j$ whenever $p_{y^*}^t \in (0, 1)$.

Next, by $e^x \ge 1 + x$ and the sign structure of $\Delta_{j,i}^t$,

$$\frac{\widetilde{D}_{y^*}}{D_{y^*}^{\mathrm{GD}}} = \sum_j w_j^{(y^*)} e^{-\eta'\tilde{\rho}^t \Delta_{j,y^*}^t} \ge 1 + \eta'\tilde{\rho}^t \sum_j w_j^{(y^*)}(-\Delta_{j,y^*}^t) \ge 1 + c_{y^*} \eta'\tilde{\rho}^t,$$

for some $c_{y^*} > 0$ whenever $p_{y^*}^t \in (0, 1)$; here $w_j^{(i)} := \beta_j^{\mathrm{GD}} e^{z_j^t} / D_i^{\mathrm{GD}}$ are positive weights.

Now use the scaling $|\rho| = \kappa\sqrt{|\eta|}$: then $\eta'\tilde{\rho}^t = \Theta(\eta^{3/2})$, whereas $e^{\pm 2C_1\eta^2} = 1 \pm O(\eta^2)$. Hence there exists $\eta_0 > 0$ (depending only on $(\boldsymbol{p}^t, \boldsymbol{H}_{\boldsymbol{z}}^t, \|\boldsymbol{g}^t\|, \mu, \kappa, L)$) such that for $0 < |\eta| \le \eta_0$,

$$D_{y^*}^{\mathrm{SAM}} \ge e^{-2C_1\eta^2} \widetilde{D}_{y^*} \ge D_{y^*}^{\mathrm{GD}}\left(1 + \tfrac{1}{2} c_{y^*} \eta'\tilde{\rho}^t\right).$$

Since $\alpha_i = \left(\sum_j e^{z_j^t}\right)/D_i$, we obtain for $0 < |\eta| \le \eta_0$:

$$\alpha_{y^*}^{\mathrm{SAM}} \le \alpha_{y^*}^{\mathrm{GD}},$$

with strict inequality under the stated nondegeneracy conditions (because then $c_{y^*} > 0$). If $\|\boldsymbol{g}^t\| = 0$ (so $\tilde{\rho}^t = 0$ by convention), equality holds. This completes the proof. $\qquad \square$

The proof of Corollary B.7 relies on the sign of $\Delta_{j,i}$, which holds automatically for $i = y^*$. However, this need not be true for an arbitrary ground-truth label $y$, except in certain special cases such as binary classification. Therefore, we further introduce the following notation and a technical assumption. Intuitively, we require that most of the probability mass concentrates on the two classes of interest, $y$ and $y^*$.

**Notation.**  We define

$$S := p_y^t + p_{y^*}^t, \qquad \tau := 1 - S, \qquad \bar{p}_y := \frac{p_y^t}{S} \in (0, 1), \qquad \Delta_{\mathrm{bin}}(\bar{p}_y) := 4\bar{p}_y(1 - \bar{p}_y)^2,$$

and let $B := \sqrt{2}$.

**Assumption B.8** (Top-2 dominance and feasible curvature gap). *Under the notation above, assume there exists $\gamma_0 > 0$ such that*

$$\frac{4B\,e^2}{p_{y^*}^t}\,\tau \;\leq\; \gamma_0 \;\leq\; \Delta_{\mathrm{bin}}(\bar{p}_y) - 6\tau.$$

**Remark.**  For any fixed $\bar{p}_y \in (0, 1)$, $\Delta_{\mathrm{bin}}(\bar{p}_y) = 4\bar{p}_y(1 - \bar{p}_y)^2$ is a positive constant, and hence $\Delta_{\mathrm{bin}}(\bar{p}_y) - 6\tau$ remains strictly positive for all sufficiently small $\tau$, while the left-hand side scales as $O(\tau)$ as $\tau \to 0$. Hence, whenever the tail mass $\tau = 1 - (p_y^t + p_{y^*}^t)$ is sufficiently small, there exists a non-empty interval of feasible $\gamma_0$ satisfying the assumption. Intuitively, this corresponds to the regime where most probability mass is concentrated on the two dominant classes $y$ and $y^*$.

**Lemma B.9** (Top-2 reduction of the curvature gap). *Under the notation above, for softmax cross-entropy,*

$$\Delta_{y^*,y}^t := (\boldsymbol{H}_{\boldsymbol{z}}^t \boldsymbol{g}^t)_{y^*} - (\boldsymbol{H}_{\boldsymbol{z}}^t \boldsymbol{g}^t)_y = \Delta_{\mathrm{bin}}(\bar{p}_y) + \zeta^t, \qquad |\zeta^t| \leq 6\tau.$$

*Proof.* Let $a := p_y^t$, $b := p_{y^*}^t$, $S := a + b = 1 - \tau$, $u := \bar{p}_y := a/S$, and

$$q := \sum_{j \notin \{y, y^*\}} (p_j^t)^2 \leq \sum_{j \notin \{y, y^*\}} p_j^t = \tau.$$

For softmax cross-entropy, we have for any $i$,

$$(\boldsymbol{H}_{\boldsymbol{z}}^t \boldsymbol{g}^t)_i = p_i^t \Big(p_i^t - y_i - \big(\sum_k (p_k^t)^2 - p_y^t\big)\Big).$$

A straightforward expansion then yields the following explicit form:

$$\Delta_{y^*,y}^t = \Delta_{\mathrm{bin}}(u) + u(S-1) + (1 - u - 2u^2)(S^2 - 1) + (4u^3 - 6u^2 + 4u - 1)(S^3 - 1) + (2u - 1)\,S\,q,$$

where $\Delta_{\mathrm{bin}}(u) = 4u(1 - u)^2$.

Using $S = 1 - \tau$, we have

$$S^2 - 1 = (1 - \tau)^2 - 1 = -2\tau + \tau^2, \qquad S^3 - 1 = (1 - \tau)^3 - 1 = -3\tau + 3\tau^2 - \tau^3.$$

Substituting these into the expansion yields

$$\Delta_{y^*,y}^t - \Delta_{\mathrm{bin}}(u) = -A(u)\tau + B(u)\tau^2 - C(u)\tau^3 + (2u - 1)\,S\,q,$$

where

$$A(u) := u + 2(1 - u - 2u^2) + 3(4u^3 - 6u^2 + 4u - 1) = 12u^3 - 22u^2 + 11u - 1,$$

$$B(u) := (1 - u - 2u^2) + 3(4u^3 - 6u^2 + 4u - 1) = 12u^3 - 20u^2 + 11u - 2 = (2u - 1)^2(3u - 2),$$

$$C(u) := 4u^3 - 6u^2 + 4u - 1 = (2u - 1)(2u^2 - 2u + 1).$$

Hence, for all $u \in [0, 1]$, the coefficient bounds satisfy

$$|A(u)| \leq 1, \qquad |B(u)| \leq 2, \qquad |C(u)| \leq 1.$$

Finally, since $\tau \in [0, 1]$ implies $\tau^2 \leq \tau$ and $\tau^3 \leq \tau$, and $Sq \leq \tau$, we obtain

$$\big|\Delta_{y^*,y}^t - \Delta_{\mathrm{bin}}(u)\big| \leq |A(u)|\tau + |B(u)|\tau^2 + |C(u)|\tau^3 + |2u - 1|Sq \leq 6\tau.$$

Setting $\zeta^t := \Delta_{y^*,y}^t - \Delta_{\mathrm{bin}}(\bar{p}_y)$ completes the proof. $\qquad\square$

**Corollary B.10** (One-step confidence ratio of $y$ under SAM vs. GD). *Under the assumptions of Theorem B.2, fix an iteration $t$ and set $\eta' = \eta\,\mu$ and $|\rho| = \kappa\sqrt{|\eta|}$. Let $y$ be the ground-truth label and $y^* = \arg\max_{j \neq y} p_j^t$. Assume the sign condition $\eta'\tilde{\rho}^t > 0$. Assume further that Assumption B.8 holds at iteration $t$.*

*Then there exists $\eta_0 = \eta_0(\boldsymbol{p}^t, \boldsymbol{H}_z^t, \|\boldsymbol{g}^t\|, \mu, \kappa, L, \gamma_0) > 0$ such that, for all step sizes $0 < |\eta| \leq \eta_0$, the following inequality holds:*

$$\alpha_y^{\mathrm{SAM}} \;\geq\; \alpha_y^{\mathrm{GD}}.$$

*Moreover, the inequality is strict whenever $\tilde{\rho}^t \neq 0$ and $p_{y^*}^t \in (0,1)$. In particular, when $\tilde{\rho}^t = 0$ (no SAM), equality holds.*

*Proof.* Let $\xi := \eta'\tilde{\rho}^t > 0$, and let $R := [V] \setminus \{y, y^*\}$. Define

$$D_y^{\mathrm{GD}} := \sum_j \beta_j^{\mathrm{GD}} e^{z_j^t}, \qquad \widetilde{D}_y := \sum_j \beta_j^{\mathrm{GD}} e^{z_j^t} \exp\{-\xi\,\Delta_{j,y}^t\}, \qquad D_y^{\mathrm{SAM}} := \sum_j \beta_j^{\mathrm{SAM}} e^{z_j^t},$$

with $\Delta_{j,i}^t := (\boldsymbol{H}_z^t\boldsymbol{g}^t)_j - (\boldsymbol{H}_z^t\boldsymbol{g}^t)_i$. By Lemma B.6, $D_y^{\mathrm{SAM}} \leq e^{2C_1\eta^2}\widetilde{D}_y$.

Introduce weights $w_j^{(y)} := \beta_j^{\mathrm{GD}} e^{z_j^t}/D_y^{\mathrm{GD}}$ so that $\sum_j w_j^{(y)} = 1$ and

$$\frac{\widetilde{D}_y}{D_y^{\mathrm{GD}}} = \sum_j w_j^{(y)} e^{-\xi\Delta_{j,y}^t} = 1 + w_{y^*}^{(y)}\big(e^{-\xi\Delta_{y^*,y}^t} - 1\big) + \sum_{j \in R} w_j^{(y)}\big(e^{-\xi\Delta_{j,y}^t} - 1\big).$$

We prove the claim in five steps.

**(1) uniform bound on $|\Delta_{j,y}^t|$.** For softmax-CE, $\|\boldsymbol{H}_z^t\|_2 \leq 1/2$ and $\|\boldsymbol{g}^t\|_2 \leq \sqrt{2}$, hence $\|\boldsymbol{H}_z^t\boldsymbol{g}^t\|_2 \leq \sqrt{2}/2$ and thus for all $i,j$,

$$|\Delta_{j,i}^t| = |(\boldsymbol{H}_z^t\boldsymbol{g}^t)_j - (\boldsymbol{H}_z^t\boldsymbol{g}^t)_i| \leq 2\|\boldsymbol{H}_z^t\boldsymbol{g}^t\|_2 \leq B, \qquad B := \sqrt{2}.$$

**(2) tail weight concentration under top-2 dominance.** Since each component of $\boldsymbol{g}^t$ lies in $[-1,1]$, we have for any $i,j$, $e^{-2|\eta'|} \leq \beta_j^{\mathrm{GD}} \leq e^{2|\eta'|}$. Therefore,

$$\sum_{j \in R} w_j^{(y)} = \frac{\sum_{j \in R}\beta_j^{\mathrm{GD}} e^{z_j^t}}{\sum_k \beta_k^{\mathrm{GD}} e^{z_k^t}} \leq \frac{e^{2|\eta'|}\sum_{j \in R} e^{z_j^t}}{e^{-2|\eta'|}\sum_k e^{z_k^t}} = e^{4|\eta'|}\sum_{j \in R} p_j^t = e^{4|\eta'|}\tau.$$

Moreover, for $|\eta'|$ sufficiently small, $w_{y^*}^{(y)} \geq \frac{1}{2}p_{y^*}^t$ (since $w_{y^*}^{(y)} = \beta_{y^*}^{\mathrm{GD}} p_{y^*}^t/\sum_k \beta_k^{\mathrm{GD}} p_k^t \geq e^{-4|\eta'|}p_{y^*}^t$ and $e^{-4|\eta'|} \geq 1/2$).

**(3) control the tail term via centering.** Using $|e^x - 1| \leq |x|e^{|x|}$ and Step (1),

$$\left|\sum_{j \in R} w_j^{(y)}\big(e^{-\xi\Delta_{j,y}^t} - 1\big)\right| \leq \sum_{j \in R} w_j^{(y)} \cdot \xi|\Delta_{j,y}^t|e^{\xi|\Delta_{j,y}^t|} \leq \xi B e^{\xi B}\sum_{j \in R} w_j^{(y)}.$$

Combining with Step (2) and choosing $\eta_0$ sufficiently small so that $\xi B \leq 1$ and $4|\eta'| \leq 1$ (which implies $e^{\xi B} \leq e$ and $e^{4|\eta'|} \leq e$), we obtain the explicit bound

$$\left|\sum_{j \in R} w_j^{(y)}\big(e^{-\xi\Delta_{j,y}^t} - 1\big)\right| \leq \xi B\, e^{\xi B}\, e^{4|\eta'|}\tau \leq \xi B e^2\,\tau.$$

By Assumption B.8 (the feasibility side), $\tau \leq \frac{p_{y^*}^t}{4Be^2}\gamma_0$. Hence

$$\xi B e^2\tau \leq \frac{\xi}{4}p_{y^*}^t\gamma_0 \leq \frac{\xi}{2}w_{y^*}^{(y)}\gamma_0,$$

where the last inequality uses $w_{y^*}^{(y)} \geq \frac{1}{2}p_{y^*}^t$.

**(4) top-2 main term and curvature gap.** By Lemma B.9, $\Delta_{y^*,y}^t \geq \Delta_{\mathrm{bin}}(\bar{p}_y) - 6\tau$. Hence Assumption B.8 implies $\Delta_{y^*,y}^t \geq \gamma_0 > 0$. Since $\xi\Delta_{y^*,y}^t \geq 0$, the second-order Taylor bound gives $e^{-u} \leq 1 - u + u^2/2$ for $u \geq 0$, hence

$$w_{y^*}^{(y)}\big(e^{-\xi\Delta_{y^*,y}^t} - 1\big) \leq -\xi w_{y^*}^{(y)}\Delta_{y^*,y}^t + \frac{\xi^2}{2}w_{y^*}^{(y)}(\Delta_{y^*,y}^t)^2 \leq -\xi w_{y^*}^{(y)}\gamma_0 + \frac{\xi^2}{2}w_{y^*}^{(y)}B^2.$$

Choose $\eta_0$ further so that $\xi \leq \frac{\gamma_0}{4}$; since $B^2 = 2$, this yields $\frac{\xi^2}{2}w_{y^*}^{(y)}B^2 \leq \frac{\xi}{4}w_{y^*}^{(y)}\gamma_0$.

**(5) combine bounds.** Putting Steps (3)–(4) together,

$$\frac{\widetilde{D}_y}{D_y^{\mathrm{GD}}} \leq 1 - \xi w_{y^*}^{(y)}\gamma_0 + \frac{\xi}{4}w_{y^*}^{(y)}\gamma_0 + \frac{\xi}{2}w_{y^*}^{(y)}\gamma_0 = 1 - \frac{\xi}{4}w_{y^*}^{(y)}\gamma_0.$$

Therefore $\widetilde{D}_y \leq D_y^{\mathrm{GD}}\big(1 - \frac{\xi}{4}w_{y^*}^{(y)}\gamma_0\big)$. Finally, using $D_y^{\mathrm{SAM}} \leq e^{2C_1\eta^2}\widetilde{D}_y$ and the scaling $\xi = \eta'\tilde{\rho}^t = \Theta(\eta^{3/2})$ while $e^{2C_1\eta^2} = 1 + O(\eta^2)$, we may further reduce $\eta_0$ so that the $O(\eta^2)$ factor is dominated by $\frac{\xi}{4}w_{y^*}^{(y)}\gamma_0$, yielding $D_y^{\mathrm{SAM}} \leq D_y^{\mathrm{GD}}$.

Since $\alpha_i = \big(\sum_j e^{z_j^t}\big)/D_i$, this implies $\alpha_y^{\mathrm{SAM}} \geq \alpha_y^{\mathrm{GD}}$. Strictness holds when $\xi > 0$, $w_{y^*}^{(y)} > 0$ and $\gamma_0 > 0$ (in particular when $\tilde{\rho}^t \neq 0$ and $p_{y^*}^t \in (0,1)$); if $\tilde{\rho}^t = 0$ then $\xi = 0$ and equality holds. $\qquad\square$

## C   Implementation

---
**Algorithm 1** Logits-SAM (one training step)

---
**Require:** model parameters $\theta$, batch $(x, y^+, y^-)$, radius $\rho$
1: $W \leftarrow$ `lm_head.weight`
2: **Forward pass:**
3:     Compute hidden states $H$ and pre-perturbation loss $\ell_{\mathrm{pre}}$
4: $g \leftarrow \nabla_W \ell_{\mathrm{pre}}$
5: $e \leftarrow \rho\, g/\|g\|$
6: **Perturbed forward pass:**
7:     `logits` $\leftarrow \mathrm{linear}(H, W + e)$
8:     Compute perturbed loss $\ell_{\mathrm{post}}$
9: **Update:**
10:     Backpropagate $\ell_{\mathrm{post}}$
11:     Optimizer step on $\theta$

---

## D   Equivalence Between Theoretical and Practical Settings

Table 5: Comparison between theoretical and practical settings of DPO with SAM. Although the signs differ for $y^-$, the resulting dynamics are equivalent. For $y^+$, the settings coincide.

| Class | Objective | Learning rate | $\rho$ | Setting |
|---|---|---|---|---|
| $y^+$ | Positive objective $f = -\log p$ | Positive ($\eta > 0$) | Positive | Theory = Practice |
| $y^-$ (Theory) | Positive objective $f = -\log p$ | Negative ($\eta < 0$) | Negative | Theory |
| $y^-$ (Practice) | Negative objective $f = \log p$ | Positive ($\eta > 0$) | Positive | Practice |

**Equivalence of sign conventions for $y^-$.** Theoretical setting ($f = -\log p$, $\eta^- < 0$, $\rho^- < 0$):

$$\boldsymbol{\theta}_{t+1}^{\mathrm{theory}} = \boldsymbol{\theta}_t - \eta^-\nabla f(\boldsymbol{\theta}_t).$$

Practical setting uses $\tilde{f} = -f$, $\tilde{\eta} = -\eta^- > 0$:

$$\boldsymbol{\theta}_{t+1}^{\mathrm{prac}} = \boldsymbol{\theta}_t - \tilde{\eta}\nabla\tilde{f}(\boldsymbol{\theta}_t) = \boldsymbol{\theta}_t - (-\eta^-)(-\nabla f(\boldsymbol{\theta}_t)) = \boldsymbol{\theta}_{t+1}^{\mathrm{theory}}.$$

| Method | Objective | Hyperparameter |
|---|---|---|
| SLiC-HF | $\max\big(0,\ \delta - \log \pi_\theta(y_w \mid x) + \log \pi_\theta(y_l \mid x)\big) - \lambda \log \pi_\theta(y_w \mid x)$ | $\lambda \in \{0.1,\ 0.5,\ 1.0,\ 10.0\};$ $\delta \in \{0.5,\ 1.0,\ 2.0,\ 10.0\}$ |
| CPO | $-\log \sigma\big(\beta \log \pi_\theta(y_w \mid x) - \beta \log \pi_\theta(y_l \mid x)\big) - \lambda \log \pi_\theta(y_w \mid x)$ | $\lambda = 1.0;\ \beta \in \{0.01,\ 0.05,\ 0.1\}$ |

Table 6: Objectives and hyperparameters for SLiC-HF and CPO.

| Method | Pythia-2.8B | Mistral-7B |
|---|---|---|
| DPO | $1 \times 10^{-4}$ | $1 \times 10^{-5}$ |
| SLiC-HF | $1 \times 10^{-3}$ | $1 \times 10^{-4}$ |
| CPO | $1 \times 10^{-4}$ | $1 \times 10^{-5}$ |

Table 7: Choice of $\rho$ for logits-SAM.

For SAM, theoretical perturbation and update:

$$\boldsymbol{\epsilon}_t^{\text{theory}} = \rho^- \frac{\nabla f(\boldsymbol{\theta}_t)}{\|\nabla f(\boldsymbol{\theta}_t)\|}, \qquad \boldsymbol{\theta}_{t+1}^{\text{theory}} = \boldsymbol{\theta}_t - \eta^- \nabla f(\boldsymbol{\theta}_t + \boldsymbol{\epsilon}_t^{\text{theory}}).$$

Practical setting uses $\tilde{f} = -f,\ \tilde{\rho} = -\rho^- > 0,\ \tilde{\eta} = -\eta^- > 0$:

$$\boldsymbol{\epsilon}_t^{\text{prac}} = \tilde{\rho} \frac{\nabla \tilde{f}(\boldsymbol{\theta}_t)}{\|\nabla \tilde{f}(\boldsymbol{\theta}_t)\|} = \rho^- \frac{\nabla f(\boldsymbol{\theta}_t)}{\|\nabla f(\boldsymbol{\theta}_t)\|} = \boldsymbol{\epsilon}_t^{\text{theory}},$$

$$\boldsymbol{\theta}_{t+1}^{\text{prac}} = \boldsymbol{\theta}_t - \tilde{\eta} \nabla \tilde{f}(\boldsymbol{\theta}_t + \boldsymbol{\epsilon}_t^{\text{prac}}) = \boldsymbol{\theta}_{t+1}^{\text{theory}}.$$

## E  ADDITIONAL EXPERIMENTAL DETAILS

**Benchmark details.**  **AlpacaEval 2** (Dubois et al., 2024) is a large-scale preference benchmark for open-ended instruction following that uses LLM-as-a-judge calibrated to human preferences; its evaluation set contains 805 single-turn instructions, and models are typically compared in pairwise settings against a baseline. **Arena-Hard v0.1** (Li et al., 2024) is a challenging subset of difficult user instructions mined from Chatbot Arena; it enables fine-grained, head-to-head comparisons between models via pairwise judging and comprises 500 hard prompts. **MT-Bench** (Zheng et al., 2023) is a multi-turn dialogue benchmark that tests a model's ability to handle diverse conversational tasks across several categories; the standard evaluation set consists of 80 multi-turn questions.

**Training details.**  For experiments on Pythia-2.8B, we use two NVIDIA A100 GPUs with data-parallel training under DDP; for Mistral-7B, we use four NVIDIA A100 GPUs with parallel training via DeepSpeed ZeRO-3 (Rasley et al., 2020).

## F  LLM USAGE STATEMENT

In preparing this manuscript, we employed an LLM as an auxiliary tool. Specifically, the LLM was used to assist with proofreading, formatting, grammar checking, and verification of mathematical proofs. All content and conclusions remain the responsibility of the authors.

