# OpenReview forum: "Sharpness-Aware Minimization in Logit Space Efficiently Enhances Direct Preference Optimization"
_ICLR.cc/2026/Conference — ICLR 2026 Poster_

### Official Review · Reviewer_vz7v · 2025-10-27

**Soundness:** 3
**Presentation:** 2
**Contribution:** 3
**Rating:** 4
**Confidence:** 4

**Summary:**

This paper investigates the "squeezing effect" in DPO, where preferred response probabilities unintentionally decrease. Through a theoretical analysis of logit space dynamics, the authors identify rapid expansion along high-curvature directions under standard optimization as the cause. They propose that Sharpness-Aware Minimization (SAM) mitigates this via curvature regularization.

To apply this efficiently, they advocate for logits-SAM, perturbing only the final layer. Empirical results on Pythia-2.8B and Mistral-7B across various alignment tasks show that logits-SAM consistently improves DPO performance with negligible computational overhead.

**Strengths:**

1. The paper tackles the important and recently identified "squeezing effect" issue in DPO, which is a practical limitation of the algorithm.
2. It provides a novel theoretical framework connecting parameter-space and logit-space dynamics, specifically analyzing the role of curvature in the squeezing effect under GD and how SAM counteracts it. The insight about matching the sign of the perturbation radius $\rho$ with the effective learning rate $\eta$ is particularly interesting.
3. Based on their theoretical insights into curvature and the squeezing effect, the paper identifies the computationally efficient logits-SAM variant as a suitable mitigation strategy and successfully applies it to the DPO context, demonstrating its practical value for improving alignment.

**Weaknesses:**

1. The core theoretical analysis simplifies DPO by modeling the dynamics associated with preferred ($y^+$, using positive $\eta$) and dispreferred ($y^-$, using negative $\eta$) responses separately within a multiclass classification framework. However, the actual DPO loss function couples the gradients for $y^+$ and $y^-$ through the sigmoid of their log-likelihood ratio difference. This theoretical separation doesn't fully capture the coupled nature of the true DPO gradient dynamics, potentially limiting the direct applicability of conclusions drawn from the simplified, independent analyses (e.g., regarding negative $\eta/\rho$).
2. A key aspect of the squeezing effect is the unintended probability increase of the most confident incorrect prediction ($y^\star$). While the theory (Corollary 3.6) and toy example (Fig 1b) suggest SAM suppresses this increase, the main real-world experiment (Fig 1c) only tracks the chosen ($y^+$) and rejected ($y^-$) probabilities. It lacks direct empirical evidence showing SAM also prevents the rise of $y^*$ probabilities during actual DPO training.

**Questions:**

1. The theoretical framework links the mitigation of the squeezing effect (when modeling the $y^-$ dynamics with a conceptual $\eta < 0$) to using a negative perturbation radius ($\rho < 0$). Since $\rho$ in standard SAM represents the magnitude of perturbation within a neighborhood, could the authors provide further intuition or clarification on the interpretation and role of a negative $\rho$ in this theoretical context, especially regarding how it achieves the desired curvature regularization?
2. Could the authors provide empirical results from the real-world DPO experiments (similar to Figure 1c) that explicitly track the probability dynamics of the most confident incorrect prediction ($y^*$)?

---

> ### Author Response · Authors · 2025-11-21
> **Response to Reviewer vz7v**
>
> We thank the reviewer for the thoughtful comments. Below, we provide point-by-point responses and hope they address the reviewer’s concerns.
>
> 1. **Simplified theoretical model vs. true coupled DPO dynamics.**
>
> This theoretical setting follows Ren & Sutherland (2025) (ICLR 2025 Best Paper), which demonstrated that the negative-gradient dynamics in DPO can be faithfully reproduced within this simplified setting, including the characteristic squeezing effect. Moreover, several phenomena observed in the multi-class logistic classification abstraction also emerge empirically during real LLM fine-tuning. These findings indicate that analyzing DPO through this abstraction provides a theoretically tractable yet practically relevant perspective on the learning behavior of large language models. We have added this discussion to Section 3.1 in the revised manuscript and further included in Section 5 a discussion of related works that study LLMs under the same or similar simplifying assumptions.
>
> Ren, Y. & Sutherland, D.J., 2025. Learning dynamics of LLM finetuning. In: Proceedings of the International Conference on Learning Representations (ICLR 2025).
>
> 2. **Missing empirical evidence for suppression of incorrect-top prediction.**
>
> We thank the reviewer for pointing out this issue, and we acknowledge that not tracking the probability of the most confident incorrect prediction in the real-world experiment (Figure 1c) was an oversight on our part. In the revised manuscript, we have added the probability trajectory of the most confident incorrect response. In addition, we include analogous curves for a larger model, Pythia-2.8B, showing the log-probabilities of the chosen, rejected, and most confident responses in Figure 1d. We hope these additions address the reviewer’s concerns.
>
> 3. **Interpretation of negative $\rho$ in theory.**
>
> For SAM, when $\rho$ is negative, the perturbation radius should be understood through its absolute value, and a positive versus negative $\rho$ corresponds respectively to performing an internal gradient ascent versus an internal gradient descent step. Since a negative $\eta$ induces gradient ascent in the outer update, choosing $\rho$ with the same sign as $\eta$ (e.g., negative $\rho$ together with negative $\eta$) leads to opposing internal and external update directions, which yields a more robust update. The specific curvature-regularization effect of a negative $\rho$ can be seen directly from Equation (5). When $\rho<0$, the update counteracts the expansion along high-curvature directions by modifying the effective growth rate from $\lambda_k$ to $\lambda_k+\rho \lambda_k^2$ (noting that $\rho<0$). We added a discussion in Section 3 regarding negative learning rates and negative $\rho$, and we provide a complete derivation and a comparative summary table in Appendix B to contrast our theoretical setup with commonly used practical configurations. We hope these additions address the reviewer’s concerns.

---

> > ### Comment · Reviewer_vz7v · 2025-11-25
> >
> > Thanks for the detailed response and the revision. The updated Figure 1 and Appendix B resolve my concerns. I have adjusted my score to 6 accordingly.

---

### Official Review · Reviewer_6UP6 · 2025-11-01

**Soundness:** 2
**Presentation:** 3
**Contribution:** 2
**Rating:** 4
**Confidence:** 4

**Summary:**

This paper seeks to address the "squeezing effect" (or "likelihood displacement") issue in DPO fine-tuning. This is a potentially harmful phenomenon, where the model performance degrades during DPO when it should be improving. The authors aim to theoretically analyze this behavior, and propose the root cause is due to interaction between the DPO objective and gradient descent. It is stated that rapid model updates expanding along high-curvature directions in the logit space explains the "squeezing effect". To remedy this, the authors propose using Sharpness-Aware Minimization (SAM), which high level seeks to minimize to "flat" surfaces. Since full SAM is too slow, the authors introduce a more practical variant, logits-SAM, which perturbs the outer-layer parameters only.

**Strengths:**

- The authors present rigorous theoretical diagnosis for a known DPO failure mode. Correlating the "squeezing effect" to "high-curvature directions" in the logit space  is a specific, actionable, and clear. The theoretical contributions and discussions are also clearly written, and presented as an elegant explanation for an existing harmful phenomenon. The clear theory to practical pipeline is also well motivated. Section 3.2 predicts how SAM should behave, Fig1-a,b confirm this prediction on a toy problem, and Fig1-c shows it on a real GPT-2 model.

- The message being delivered is also succinct: choose the SAM hyperparameter with the same sign as the learning rate to alleviate the squeezing effect.

- The paper's primary practical contribution is the logits-SAM algorithm. This identifies that only the last layer needs perturbation, and the authors propose a simple practical "fix". Beyond just merely stating the instability of DPO and analyzing its failure modes, this additional practical solution is a strong addition.

**Weaknesses:**

- This method introduces a new highly sensitive hyperparameter ($\rho$). Table 3 demonstrates that while $\rho=10^{-4}$ might yield good results, the slightly larger $\rho=10^{-3}$ actually performs worse than the baseline. The optimal range seems extremely small. The majority of preference fine-tuning methods already encompass highly sensitive hyperparameters, and discussions about reference-based versus reference-free preference fine-tuning, in order to shift towards more robust solutions that are cheaper to tune.

- Logits-SAM is not a new method. There are several works in this area recently, making this paper's contribution mildly incremental. The novelty of this work is in its application of logits-SAM in this particular setting. [1], [2], [3]

- Efficiency is highly model-dependent. The claim that overhead is negligible requires the final output later to be a tiny fraction of the total model parameters. This is true for many decoder-only models, but it is not a universal guarantee. Hence the experiments conducted are all on small models like GPT-2 with toy datasets, and are largely outdated.

[1] Wen, Kaiyue, Tengyu Ma, and Zhiyuan Li. "How does sharpness-aware minimization minimize sharpness?." arXiv preprint arXiv:2211.05729 (2022).

[2] Li, Dongqi, et al. "Sharpness-aware minimization for out-of-distribution generalization." International Conference on Neural Information Processing. Singapore: Springer Nature Singapore, 2023.

[3] Foret, Pierre, et al. "Sharpness-aware minimization for efficiently improving generalization." arXiv preprint arXiv:2010.01412 (2020).

**Questions:**

- Section 3.2 explicitly argues that for the negative learning rate in DPO, "one should choose a negative $\rho$". Section 3.3 says that the implementation "consistently use a positive $\rho$". The claim that this is equivalent hold for first-order methods, but is there a leap to assume it holds for curvature-aware methods like SAM?

- Have the authors conducted experiments on larger more currently relevant models beside GPT-2?

- DPO (Eq. 2) already has an implicit KL regularization term. AdamW has weight decay. How does SAM's curvature regularization interact with these forms of regularization?

- Why is perturbing only the last layer sufficient? Proposition 3.1 links the full parameter Hessian to the logit Hessian via a features map. Yet this is the output of all other layers. Can the authors give any intuition on why this is sufficient to regularize the entire system? Or is GPT-2 the limitation?

---

> ### Author Response · Authors · 2025-11-21
> **Response to Reviewer 6UP6 (part 1/2)**
>
> We thank the reviewer for the insightful and detailed comments. Before addressing the points individually, we would like to clarify an important misunderstanding: only Figure 1(c) uses GPT-2, whereas all experiments in Section 4 are conducted on modern billion-parameter models, including Pythia-2.8B, Mistral-7B, and Gemma-2B-IT, and follow standard evaluation protocols for DPO-style algorithms. Therefore, the use of GPT-2 is not a limitation of our work. Below, we provide point-by-point responses and hope they address the reviewer’s concerns.
>
> 1. **Hyperparameter sensitivity.**
>
> As the reviewer noted, preference fine-tuning methods for LLMs generally involve highly sensitive hyperparameters. However, we do not believe that the $\rho$ hyperparameter in logits-SAM falls into this category. For example, the learning rate in DPO can significantly affect final performance when changed from $2\times 10^{-6}$ to $3\times 10^{-6}$, whereas in our sensitivity analysis we intentionally varied $\rho$ over a very wide range from $1\times 10^{-5}$ to $1\times 10^{-2}$, spanning three orders of magnitude. The performance drop at large $\rho$ values therefore does not indicate that $\rho$ is inherently more sensitive than typical hyperparameters used in preference optimization. We also note that the degradation at excessively large $\rho$ values is a well-known behavior shared by all SAM-style methods rather than being specific to logits-SAM, as also observed in prior work such as [4] and [5].
>
> To further alleviate the reviewer’s concern, we added to the revised manuscript (Section 4.3, Figure 3) the learning curves of all $\rho$ choices for Mistral-7B, which clearly show that logits-SAM consistently outperforms the baseline across the full search range.
>
> 2. **Limited novelty of logits-SAM.**
>
> We would like to clarify that the methods in [1], [2], and [3] do not incorporate logits-SAM. In particular, these works proposed and analyzed SAM and its theoretical properties in a general context, mostly for classification. In contrast, following the theoretical setting introduced in [8] (ICLR 2025 Best Paper Award), we theoretically and empirically demonstrate that logits-SAM can mitigate the squeezing effect of DPO, thereby improving DPO’s performance in LLM alignment. Moreover, our **theoretical contributions** are **entirely different** from those presented in [1], [2], and [3]. Therefore, we **strongly believe** that the novelty of our research is not affected by [1], [2], and [3].
>
> Additionally, some works, such as [6] and [7], do mention it, and we agree that logits-SAM itself is not a new algorithm, as discussed in our manuscript. However, this does not imply that our contribution lacks novelty. In [6] and [7], logits-SAM appears only as a byproduct of their analysis and is largely overlooked. To the best of our knowledge, logits-SAM has not been adopted in supervised learning settings such as image classification, as it generally performs poorly there. In contrast, within the context of DPO, we provide a theoretical explanation showing that logits-SAM mitigates the squeezing effect, and we validate this insight through extensive experiments. According to the spirit of the No Free Lunch Theorem, the effectiveness of an algorithm fundamentally depends on the task at hand. Our contribution lies in recognizing that this previously overlooked algorithm is in fact well-suited for DPO.
>
> 3. **Efficiency claim is model-dependent.**
>
> We agree that the additional computational overhead is model-dependent. However, we observe that for deeper models, the output layer typically constitutes an even smaller fraction of the total parameters (a useful analogy is a tall and narrow cake). For example, in the models used in our experiments, the output layer accounts for 4.64% of the parameters in Pythia-2.8B and 1.81% in Mistral-7B, as reported at the end of Section 3. For larger models such as LLaMA2-70B, this ratio further decreases to 0.37%.
>
> 4. **Sign of learning rate under SAM.**
>
> Although SAM induces an implicit curvature regularization effect, it is fundamentally still a first-order method because its update does not use second-order information. Under a fixed perturbation, applying a positive learning rate to the negative objective is therefore equivalent to applying a negative learning rate to the positive objective. What remains is to account for the sign of $\rho$. Combining these observations, we show that “negative objective + positive learning rate + positive $\rho$” is equivalent to “positive objective + negative learning rate + negative $\rho$.” While this relationship may seem unintuitive at first glance, we provide a complete derivation and a summary of this equivalence in Appendix B of the revised manuscript.

---

> ### Author Response · Authors · 2025-11-21
> **Response to Reviewer 6UP6 (part 2/2)**
>
> 5. **Experiments on larger and modern models.**
>
>  As noted at the beginning of our response, the initial version of our paper already included experiments on Pythia-2.8B and Mistral-7B. In addition, during the rebuttal period, we further explored applying logits-SAM in a different setting. Specifically, we evaluated the methods on the AI-safety benchmark SorryBench under an on-policy setting, and the results are shown below. We observe that integrating logits-SAM leads to substantial performance gains: DPO with logits-SAM avoids the degradation in refusal rate and performs better than the reference model. Moreover, combining logits-SAM with the CHES method of [10] further increases the refusal rate, yielding an absolute improvement of approximately 9% on both the training and test sets. These findings demonstrate that logits-SAM can be effectively transferred to other settings and tasks. We have incorporated this new experiment into the revised manuscript, as shown in Table 4 in Section 4.3.
>
> **Table: Train and test refusal rates for different methods on SorryBench (higher is better).**
>
> | Method            | Ref model | DPO    | DPO + logits-SAM | CHES [10] | CHES + logits-SAM |
> |-------------------|-----------|--------|-------------------|-----------|--------------------|
> | **Train Refusal** | 0.8054    | 0.7703 | **0.8135**        | 0.8459    | **0.9324**         |
> | **Test Refusal**  | 0.7231    | 0.7077 | **0.7538**        | 0.7846    | **0.8769**         |
>
> 6. **Interaction with KL regularization and weight decay.**
>
> Although we do not provide a rigorous theoretical analysis in this regard, we can offer some intuitive explanations. The KL regularization in DPO encourages the fine-tuned model to stay close to the pretrained model, and SAM helps by searching for flatter regions in the neighborhood of the pretrained parameter initialization. As for AdamW, the weight decay term is typically implemented as part of the optimizer’s update rule rather than being included in the gradient computation itself, and therefore it does not affect the perturbation used by SAM.
>
> 7. **Why perturbing only the last layer is sufficient.**
>
> Prior work [8] and [9], which use billion-parameter models, has shown that the dynamics of the output layer often dominate the overall update behavior during LLM fine-tuning. This provides further justification for focusing the perturbation on this layer and may explain why logits-SAM remains effective even when perturbing only the final layer. In addition, the original SAM algorithm is difficult to apply in the LLM setting due to its substantial computational overhead, whereas our experiments demonstrate that applying SAM only to the output layer is sufficient to achieve significant performance gains. This yields a favorable trade-off between efficiency and effectiveness.
>
> [4] Andriushchenko, M., & Flammarion, N. (2022). Towards understanding sharpness-aware minimization. In Proceedings of the International Conference on Machine Learning (pp. 639–668). PMLR.
>
> [5] Liu, Y., Mai, S., Cheng, M., Chen, X., Hsieh, C.-J., & You, Y. (2022). Random sharpness-aware minimization. Advances in Neural Information Processing Systems, 35, 24543–24556.
>
> [6] Christina Baek, Zico Kolter, and Aditi Raghunathan. Why is sam robust to label noise? arXiv preprint arXiv:2405.03676, 2024.
>
> [7] Sidak Pal Singh, Hossein Mobahi, Atish Agarwala, and Yann Dauphin. Avoiding spurious sharpness minimization broadens applicability of sam. arXiv preprint arXiv:2502.02407, 2025.
>
> [8] Yi Ren and Danica J Sutherland. Learning dynamics of llm finetuning. arXiv preprint arXiv:2407.10490, 2024.
>
> [9] Sadhika Malladi, Alexander Wettig, Dingli Yu, Danqi Chen, and Sanjeev Arora. A kernel-based view of language model fine-tuning. In International Conference on Machine Learning, pp.23610–23641. PMLR, 2023.
>
> [10] Noam Razin, Sadhika Malladi, Adithya Bhaskar, Danqi Chen, Sanjeev Arora, and Boris Hanin. Unintentional unalignment: Likelihood displacement in direct preference optimization. arXiv preprint arXiv:2410.08847, 2024.

---

### Official Review · Reviewer_5zL9 · 2025-11-01

**Soundness:** 3
**Presentation:** 2
**Contribution:** 3
**Rating:** 6
**Confidence:** 3

**Summary:**

The paper analyzes the ``squeezing'' effect in DPO—where preferred responses lose likelihood—by deriving a unified logit-space dynamic that links GD and SAM and reveals high-curvature modes as the culprit. It proves that SAM, applied with a coefficient sharing the sign of the effective learning rate, damps these unstable modes. Leveraging this, **logits-SAM** perturbs only the output layer, preserving curvature-aware regularization with ~2–3% overhead. Experiments on Pythia-2.8B and Mistral-7B across HH, TL;DR, UltraFeedback, AlpacaEval 2, Arena-Hard, and MT-Bench show consistent gains and reduced sharpness, with robust performance for small ρ (≈1e-5–1e-4).

**Strengths:**

1. **Clear Theoretical Contribution:** Provides a unified logit-space dynamical analysis that links GD and SAM, pinpointing high-curvature mode amplification as the mechanism behind DPO ``squeezing,'' and proving sign-aligned SAM mitigates it.

2. **Practical, Efficient Method:** Introduces logits-SAM, an output-layer perturbation that retains curvature-aware regularization with minimal overhead (~2–3%) and seamless integration into existing DPO/SLiC-HF/CPO pipelines.

**Weaknesses:**

1. **Theory–practice gap:** Core analysis relies on first/second-order approximations in logit space (fixed features, softmax CE), which may not fully capture nonlinearity and parameter coupling in deep, decoder-only LMs.

2. **Final-layer perturbation bias:** Restricting SAM to the output layer improves efficiency but may miss sharp directions arising in earlier blocks/attention layers, potentially undercutting robustness on harder distributions.

3. **DPO-specific framing:** The mitigation is analyzed through the ``negative learning rate / squeezing'' view of DPO; it is unclear how strongly the guarantees transfer to alternative preference-learning objectives or online RLHF settings.

**Questions:**

see weakness

---

> ### Author Response · Authors · 2025-11-21
> **Response to Reviewer 5zL9 (part 1/2)**
>
> We thank the reviewer for the insightful comments. Below, we provide point-by-point responses and hope they address the reviewer’s concerns.
> 1. **Theory–practice gap.**
>
> This theoretical setting follows [1] (ICLR 2025 Best Paper), which demonstrated that the negative-gradient dynamics in DPO can be faithfully reproduced within this simplified setting, including the characteristic squeezing effect. Moreover, several phenomena observed in the multi-class logistic classification abstraction also emerge empirically during real LLM fine-tuning. These findings indicate that analyzing DPO through this abstraction provides a theoretically tractable yet practically relevant perspective on the learning behavior of large language models. We have added this discussion to Section 3.1 in the revised manuscript and further included in Section 5 a discussion of related works that study LLMs under the same or similar simplifying assumptions.
>
> 2. **Final-layer perturbation bias.**
>
> We agree with the reviewer that perturbing only the final layer may overlook sharp or unstable directions arising in earlier blocks or attention layers. However, prior work [1] and [2] has shown that, during LLM fine-tuning, the dynamics of the output layer often dominate the overall update behavior, which provides further justification for focusing the perturbation on this layer.
>
> Moreover, the original SAM algorithm doubles the computational cost and requires substantial additional memory (a quantitative analysis is provided in Section 4.3), which makes it impractical for large-scale LLM training and fine-tuning. In contrast, our simplified variant, logits-SAM, preserves performance gains while introducing almost no extra training overhead, and it is supported by a clear theoretical justification. We have validated this trade-off across multiple billion-parameter models and various datasets and benchmarks. These results show that logits-SAM strikes an effective balance between computational efficiency and performance.
>
> 3. **DPO-specific framing.**
>
> We would like to clarify that our analysis focuses on a phenomenon specific to DPO training, namely the squeezing effect, and we provide a theoretical explanation for it. However, this does not mean that our theory is limited to the DPO objective. Other preference optimization methods such as IPO [3] introduce negative-gradient updates through pairwise preference comparisons, while GRPO [4] also produces negative updates via relative disadvantages computed from reward signals. Consequently, they exhibit similar negative-gradient updates to those analyzed in our work. In principle, our theoretical framework can therefore extend to a broader class of preference-learning algorithms rather than DPO alone. That said, it remains unclear whether these alternative methods demonstrate a squeezing effect in practice, and additional motivation may be needed to justify applying our analysis to them, although the extension is technically feasible.
>
> On the other hand, to demonstrate the use of logits-SAM in an on-policy setting, we followed the experimental design in [5]. Specifically, we evaluated the methods on the AI-safety benchmark SorryBench under an on-policy setting, and the results are shown below. We observe that integrating logits-SAM leads to substantial performance gains: DPO with logits-SAM avoids the degradation in refusal rate and performs better than the reference model. Moreover, combining logits-SAM with the CHES method of [5] further increases the refusal rate, yielding an absolute improvement of approximately 9% on both the training and test sets. These findings demonstrate that logits-SAM can be effectively transferred to other settings and tasks. We have incorporated this new experiment into the revised manuscript, as shown in Table 4 in Section 4.3.
>
> **Table: Train and test refusal rates for different methods on SorryBench (higher is better).**
>
> | Method           | Ref model | DPO    | DPO + logits-SAM | CHES [5]  | CHES + logits-SAM |
> |------------------|-----------|--------|-------------------|--------|--------------------|
> | **Train Refusal** | 0.8054    | 0.7703 | **0.8135**        | 0.8459 | **0.9324**         |
> | **Test Refusal**  | 0.7231    | 0.7077 | **0.7538**        | 0.7846 | **0.8769**         |

---

> ### Author Response · Authors · 2025-11-21
> **Response to Reviewer 5zL9 (part 2/2)**
>
> [1] Yi Ren and Danica J Sutherland. Learning dynamics of llm finetuning. arXiv preprint
> arXiv:2407.10490, 2024.
>
> [2] Sadhika Malladi, Alexander Wettig, Dingli Yu, Danqi Chen, and Sanjeev Arora. A kernel-based view of language model fine-tuning. In International Conference on Machine Learning, pp.23610–23641. PMLR, 2023.
>
> [3] Azar, M. G., Rowland, M., Piot, B, Guo, D., Calandriello, D., Valko, M., & Munos, R. (2023). A General Theoretical Paradigm to Understand Learning from Human Preferences. arXiv preprint arXiv:2310.12036.
>
> [4] Shao, Z., Wang, P., Zhu, Q., Xu, R., Song, J., Bi, X., Zhang, H., Zhang, M., Li, Y. K., Wu, Y., & Guo, D. (2024). DeepSeekMath: Pushing the Limits of Mathematical Reasoning in Open Language Models. arXiv:2402.03300.
>
> [5] Noam Razin, Sadhika Malladi, Adithya Bhaskar, Danqi Chen, Sanjeev Arora, and Boris Hanin.
> Unintentional unalignment: Likelihood displacement in direct preference optimization. arXiv
> preprint arXiv:2410.08847, 2024.

---

> > ### Comment · Reviewer_5zL9 · 2025-11-24
> >
> > The SorryBench results are convincing—the improved refusal rates address my concerns about on-policy robustness.
> > On the theory-practice gap: yes, the model is simplified, but given the added discussion in 3.1 and the efficiency gains, I think the logits-SAM approach is justified for practical purposes.
> > The revisions strengthen the paper. Keeping my positive score. Good luck!

---

### Official Review · Reviewer_DsoC · 2025-11-01

**Soundness:** 3
**Presentation:** 3
**Contribution:** 3
**Rating:** 6
**Confidence:** 3

**Summary:**

This paper aims to address the squeezing effect of DPO and proposes a new DPO method with leveraging sharpness-aware minimization (SAM). Comprehensive theoretical and experimental analyses have been conducted to demonstrate the effectiveness of the SAM. As such, while this work has some limitations. I still recommend an acceptance.

**Strengths:**

This work has the following strengths:

1.	This work studies on an important problem.
2.	This work proposes to leverage SAM to mitigate the squeezing effect of DPO, with providing comprehensive theoretical evidences.
3.	Extensive experiments on real-world datasets have been conducted to verify the efficacy of the proposed method.

**Weaknesses:**

I also have some concerns:

1.	On the description of “gradient descent with a negative learning rate”: The phrase “gradient descent with a negative learning rate” is unconventional, as learning rates in DPO are typically positive. This may cause unnecessary confusion for readers. From the theoretical analysis, I understand that DPO may, in certain cases, apply a reversed gradient direction. However, it would be more precise to present this as theoretically equivalent to using a negative learning rate, rather than stating that the learning rate is negative outright. This clarification is important to prevent misinterpretation.

2.	Gap between theoretical setting and practical application: For ease of theoretical derivation, the authors formulate DPO as a multi-class logistic classification problem. While this abstraction facilitates analysis, it may not fully reflect practical usage scenarios of DPO. Additional explanation should be provided to justify the relevance and applicability of the theoretical setting to real-world DPO implementations.

3.	Many techniques in the theoretical section appear to follow the framework introduced by Ren & Sutherland. This is not necessarily a critical limitation, as research naturally builds upon prior work. However, the following questions should be addressed: (1) What are the specific theoretical challenges in integrating SAM into DPO? (2) What are the novel theoretical contributions of this work compared with Ren & Sutherland’s approach?



4.	I also some concerns on the experiments: 1) The “squeezing effect” is not a new concept and has been studied in recent work (e.g., [a1], [a2]). How does the proposed method perform compared with these recent baselines? 2) It would be better to include more experiments on diverse backbones and benchmark.

[a1] C2-DPO: Constrained Controlled Direct Preference Optimization (arxiv'25)
[a2] Unintentional unalignment: Likelihood displacement in direct preference optimization (ICLR’25)

**Questions:**

Please refer to weaknesses.

---

> ### Author Response · Authors · 2025-11-21
> **Response to Reviewer DsoC**
>
> We thank the reviewer for the thoughtful and detailed feedback. Below, we provide point-by-point responses and hope they effectively address the reviewer’s concerns.
> 1. **Terminology clarity (negative learning rate).**
>
> We agree that the phrase “gradient descent with a negative learning rate” is unconventional and may lead to misunderstandings. In the revised manuscript, we have replaced it with “negative-gradient updates” and explicitly clarified in the Introduction that the two formulations are algorithmically equivalent, thereby eliminating potential confusion.
>
> 2. **Theoretical–practical gap.**
>
> This theoretical setting follows Ren & Sutherland (2024) (ICLR 2025 Best Paper), which demonstrated that the negative-gradient dynamics in DPO can be faithfully reproduced within this simplified setting, including the characteristic squeezing effect. Moreover, several phenomena observed in the multi-class logistic classification abstraction also emerge empirically during real LLM fine-tuning. These findings indicate that analyzing DPO through this abstraction provides a theoretically tractable yet practically relevant perspective on the learning behavior of large language models. We have added this discussion to *Section 3.1* in the revised manuscript and further included in *Section 5* a discussion of related works that study LLMs under the same or similar simplifying assumptions.
>
> 3. **Difference from Ren & Sutherland (2024).**
>
> We would like to clarify that although we adopt the same setting as Ren & Sutherland (2024), namely multi-class logistic classification with fixed features, the theoretical framework and proof techniques used in our work are fundamentally different. Our analysis primarily focuses on the geometric structure in logit space (Propositions 3.1 and 3.3) and on the dynamics induced by the Hessian (Theorem 3.2). These components together lead to the conclusion that directions of high curvature expand rapidly (Corollary 3.4). In comparison, the work of Ren & Sutherland (2024) does not work on curvature or Hessian-based arguments. In addition, our decomposition in the eigenbasis of the logit Hessian is also novel and provides a concise and intuitive scalar equation that highlights the effect of SAM. Taken together, our theoretical framework is more general and may be extended to broader settings by considering different pullbacks.
>
> A key technical challenge in integrating SAM into DPO is that the error terms contain remainder terms that depend on both $\eta$ and $\rho$. Addressing this difficulty requires careful control of these terms, and the geometric structure we analyze makes it possible to handle them in a clean and tractable manner.
>
> 4. **Experimental concerns (squeezing effect and additional baselines).**
>
> We thank the reviewer for pointing out additional studies on the squeezing effect. Our initial manuscript already included [a2], and in the revised version we have expanded the discussion of related work on this topic, including both [a1] and [a2], in Section 5. It is worth noting that, unlike existing approaches that modify the objective function or filter the training data, our method adopts a purely optimization-based perspective. It is therefore conceptually orthogonal to these techniques and can be combined with them.
>
> Regarding [a1], since it is still a preprint and its code has not been released, it is difficult for us to perform a direct empirical comparison. For [a2], we conducted experiments during the rebuttal period using their publicly available implementation. We evaluated the methods on the AI-safety benchmark SorryBench under an on-policy setting, and the results are shown below. We observe that integrating logits-SAM leads to substantial performance gains: DPO with logits-SAM avoids the degradation in refusal rate and performs better than the reference model. Moreover, combining logits-SAM with the CHES method of [a2] further increases the refusal rate, yielding an absolute improvement of approximately 9% on both the training and test sets. These findings demonstrate that logits-SAM can be effectively transferred to other settings and tasks. We have incorporated this new experiment into the revised manuscript, as shown in Table 4 in Section 4.3.
>
> **Table: Train and test refusal rates for different methods on SorryBench (higher is better).**
>
> | Method           | Ref model | DPO    | DPO + logits-SAM | CHES [a2]  | CHES + logits-SAM |
> |------------------|-----------|--------|-------------------|--------|--------------------|
> | **Train Refusal** | 0.8054    | 0.7703 | **0.8135**        | 0.8459 | **0.9324**         |
> | **Test Refusal**  | 0.7231    | 0.7077 | **0.7538**        | 0.7846 | **0.8769**         |

---

### Author Response · Authors · 2025-11-21
**Summary of Changes in the Revised Manuscript**

We sincerely thank all reviewers for their valuable comments. We have revised the manuscript accordingly, which has substantially improved its clarity, exposition, and overall solidity. Below is a summary of the major changes:

1. **Clarifications on negative learning-rate updates.**

Following Reviewer **DsoC**’s suggestion, we revised our description of “gradient descent with a negative learning rate” to “negative-gradient updates”, and we now explicitly clarify in the *Introduction* that the two formulations are algorithmically equivalent, thereby avoiding potential misunderstandings.

In line with comments from Reviewers **DsoC**, **6UP6**, and **vz7v**, we further added a discussion in *Section 3* regarding negative learning rates and negative $\rho$, and we provide a complete derivation and a comparative summary table in *Appendix B* to contrast our theoretical setup with commonly used practical configurations.

2. **Real-world relevance of the simplified setting.**

We enriched the discussion in *Section 3.1* (as suggested by Reviewers **DsoC**, **5zL9**, **6UP6**, and **vz7v**) to better explain why the simplified setting can reflect the dynamics of real-world LLM fine-tuning, and we additionally included in *Section 5* a discussion of related works that study LLMs under the same or similar simplifying assumptions.


3. **Additional experiments on the AI-safety benchmark SorryBench under an on-policy setting.**

Based on feedback from Reviewers **DsoC**, **5zL9**, and **6UP6**, we added an experiment on the AI-safety benchmark SorryBench under an on-policy setting (*Table 4 in Section 4.3*) to demonstrate the robustness of logits-SAM when applied to different settings. The results show substantial performance improvements, with an absolute gain of approximately 9%.

4. **Expanded discussion on contemporaneous work studying the squeezing effect.**

As suggested by Reviewer **DsoC**, we added further discussion of concurrent research on the squeezing effect in *Section 5*.

5. **Additional analysis and experiments for Figure 1.**

Following Reviewers **6UP6** and **vz7v**, we supplemented the experiments in Figure 1 by adding the probability changes of the model’s most confident response, and we included corresponding results from a larger model (Pythia-2.8B) to further validate our theory.

6. **Learning curves under different $\rho$ values in Figure 3.**

Following Reviewer **6UP6**’s suggestion, we added learning curves for logits-SAM under different $\rho$ values in *Figure 3* to further illustrate the robustness of our method with respect to hyperparameter choices.

All changes are highlighted in **blue**. We encourage reviewers to take a look at the revised manuscript. We appreciate your time and valuable feedback, which have helped us substantially improve the paper. Below, we provide point-by-point responses to the specific concerns raised by each reviewer.

---

### Author Response · Authors · 2025-12-02
**Rebuttal Summary for the Area Chair**

Thank you for taking the time under the special circumstances. We provide here a concise summary of the key rebuttal clarifications to help the AC quickly grasp the main points.

In the initial reviews, the reviewers identified several shared strengths of our work, commenting on the importance of the problem, the clarity and novelty of the theory, and the thoroughness of the experiments, while also raising several questions. We responded to each of those concerns in detail during the rebuttal. Among the four reviewers, two provided replies before the leak incident: **Reviewer 5zL9** confirmed that all of their concerns had been fully addressed, noting that the additional experiments on SorryBench were convincing and that the new theory–practice discussion was appropriate, and therefore maintained their positive assessment (score of 6). **Reviewer vz7v** likewise agreed that our updates resolved their concerns and accordingly increased their score (from 4 to 6), noting that the updated experiments (Figure 1) and the additional discussion provided in Appendix B were helpful.

Although the remaining two reviewers were not able to provide responses before the incident, we believe their concerns were also thoroughly addressed. Following **Reviewer DsoC**’s suggestions, we improved the phrasing in the manuscript and elaborated on how our theoretical setup relates to practical DPO behavior. We also clarified the fundamental differences between our work and that of Ren & Sutherland (2024), and added new experiments under an additional setting (SorryBench) as requested. **Reviewer 6UP6**’s main concern was whether our method was limited to GPT-2 models; we clarified this misunderstanding by showing that our experiments span multiple billion-parameter models under various settings. To further address their conceptual questions, we provided a detailed comparison with prior work to clarify the novelty of logits-SAM. In addition, the reviewer raised several experiment-related issues, which we addressed point by point, including hyperparameter sensitivity, algorithmic efficiency, interactions with regularization, and other implementation-specific details.

We hope this summary is helpful, and we sincerely appreciate the AC’s effort under the unusual circumstances.

---

### Meta-Review · Area_Chair_6Kii · 2026-01-08

**Summary:**

The main concerns affecting the decision:
- theory–practice gap: gap from simplified logit-space approximations versus the coupled DPO gradient. (5zL9, vz7v)
- limited novelty: positioning relative to recent squeezing-effect (C2-DPO) and SAM/logits-SAM related work. (6UP6, DsoC)
- unclear details: hyperparameter sensitivity and clarity around negative learning rate. (DsoC, 6UP6, vz7v)
- breadth/robustness (whether last-layer-only perturbation suffices; whether improvements generalize across objectives/backbones and harder distributions). (DsoC, 5zL9, 6UP6)

**Reviewer Concerns:**

Addressed by the rebuttal:
- robustness: additional experiments on SorryBench were added. (5zL9)
- unclear details/missing evidence: details of hyperparameter sensitivity and clarity around negative learning rate were reported. (vz7v, DsoC)
- theory–practice gap: partially alleviated with added discussion (5zL9, vz7v, 6UP6)
- limited novelty/positioning vs recent related work: a detailed comparison with prior work was added (DsoC, 6UP6)
- breadth/robustness of experiments across diverse backbones/objectives: additional experiments were added (DsoC, 5zL9, 6UP6)

Still outstanding / partially addressed (should be framed as limitations)
- none

**Reviewer Scores:**

- DsoC (original score 6): likely no change (stays positive).
- 5zL9 (original score 6): no change (explicitly keeps positive score).
- 6UP6 (original score 4): likely small positive update
- vz7v (original score 4): score raised to 6 (already reported)

---

### Decision · Program_Chairs · 2026-01-26

Accept (Poster)